# A generative model of the hippocampal formation trained with theta driven local learning rules

**Tom M George**[1]

**Caswell Barry**[2]    **Kimberly Stachenfeld**[3,4]    **Claudia Clopath**[5,1]    **Tomoki Fukai**[6]

[1]Sainsbury Wellcome Centre, UCL, UK    [2]Dept. of Cell and Developmental Biology, UCL, UK
[3]Google DeepMind, London, UK    [4]Columbia University, New York, NY
[5]Bioengineering Dept., Imperial College, UK    [6]Okinawa Institute of Science and Technology, Japan

`tom.george.20@ucl.ac.uk`

## Abstract

Advances in generative models have recently revolutionised machine learning. Meanwhile, in neuroscience, generative models have long been thought fundamental to animal intelligence. Understanding the biological mechanisms that support these processes promises to shed light on the relationship between biological and artificial intelligence. In animals, the hippocampal formation is thought to learn and use a generative model to support its role in spatial and non-spatial memory. Here we introduce a biologically plausible model of the hippocampal formation tantamount to a Helmholtz machine that we apply to a temporal stream of inputs. A novel component of our model is that fast theta-band oscillations (5-10 Hz) gate the direction of information flow throughout the network, training it akin to a high-frequency wake-sleep algorithm. Our model accurately infers the latent state of high-dimensional sensory environments and generates realistic sensory predictions. Furthermore, it can learn to path integrate by developing a ring attractor connectivity structure matching previous theoretical proposals and flexibly transfer this structure between environments. Whereas many models trade-off biological plausibility with generality, our model captures a variety of hippocampal cognitive functions under one biologically plausible local learning rule.

## 1   Introduction

Generative models seek to create new data samples which are similar to those from the training set. To do so they must learn the probability distribution of the training data, comprising a rich, generalisable and accurate model of the world. Many of the recent advances in AI have involved types of generative models: VAEs [1], GANs [2], diffusion models [3] and autoregressive models [4] have seeded improvements in AI capabilities ranging from data compression [5] to image generation [6] and natural language [7]. In neuroscience, the animal brain has long been known to exploit generative models [8, 9]. The ability to generate representative sensory data samples can be used directly, for example during offline planning or memory recall. It can also be used indirectly to aid training of inference networks with the goal of processing rich, noisy and high dimensional streams of incoming sensory stimuli, as discussed in the predictive coding literature [10]. In a sentence: "What I cannot create [generate], I do not understand [inference]" (R. Feynman).

37th Conference on Neural Information Processing Systems (NeurIPS 2023).

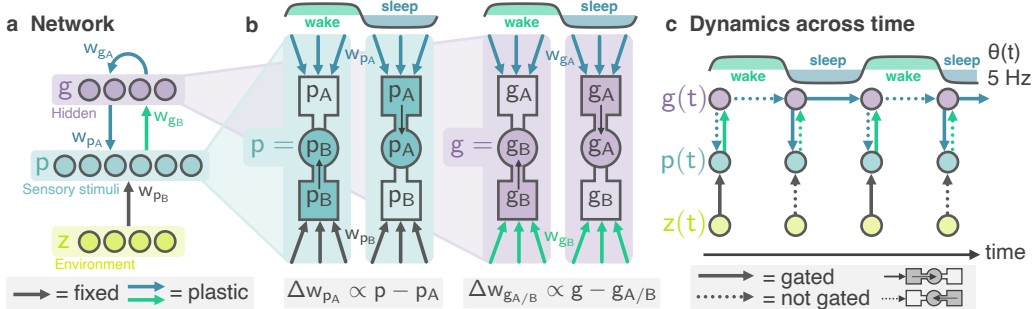

Figure 1: A biologically plausible generative model is trained with theta frequency wake-sleep cycles and a local learning rule. **a** Network schematic: high-D stimuli from an underlying environmental latent, $z$, arrive at the basal dendrites of the sensory layer, $p$, and map to the hidden layer, $g$ (this is the inference model, weights in green). Simultaneously, top-down predictions from the hidden layer $g$ arrive at the apical dendrites of $p$ (this is the generative model, weights in blue). **b** Neurons in layers $p$ and $g$ have three compartments. A fast oscillation, $\theta(t)$, gates which dendritic compartment – basal ($p_B$, $g_B$) or apical ($p_A$, $g_A$) – drives the soma. A local learning rule adjusts input weights to minimise the prediction error between dendritic compartments and the soma. **c** This equates to rapidly switching "wake" and "sleep" cycles which train the generative and inference models. Panel c displays just two updates per theta-cycle, in reality there are many ($\delta t << T_\theta$).

The hippocampal-entorhinal system (aka. hippocampal formation) – a brain structure implicated in spatial [11] and non-spatial [12] memory – provides a pertinent example. Its primary role seems to be inference [13]: mapping sensory inputs into a robust and decodable representation of state (grid cells [14], place cells [11] etc. [15]). A generative model is thought to have a dual role in learning: supporting offline tasks such as route planning [16] and memory consolidation [17], and online during behaviour with path integration [18]. Path integration enables the hippocampal network to maintain an up-to-date and accurate estimate of its position in the absence of reliable sensory data by integrating self-motion cues. A recent flurry of computational [19, 20, 21] and theoretical [22, 21] work has highlighted the importance of path integration as a key objective explaining hippocampal function and representations.

Existing computational generative models of the hippocampal formation [23, 24] account for many of its cognitive functions and internal representations but require non-trivial learning rules and message passing protocols which don't connect with known aspects of biology. Computational models of path integration [25, 26, 27] have mostly focussed on continuous attractor networks which, although experimentally supported [28], alone lack the complexity or expressivity required of a fully general model of the hippocampal memory system.

The primary contribution of this paper is to introduce a biologically plausible model of sequence learning in the hippocampus which unifies its capacities as a generative model of sensory stimuli and path integration under one schema. To do this we propose modeling the hippocampal formation as a Helmholtz machine [29] which learns to predict sensory stimuli given the current hidden state and action (e.g. velocity). We propose a deep connection between the hippocampal theta oscillation [30] and the unsupervised wake-sleep algorithm [31] for training Helmholtz machines. Though this class of generative models isn't widely used, and lacks the scalability of the lastest transformer-based sequence learners, it excels in this context since is has many natural points of contact with biology (both in terms of architecture and neural dynamics) yet still maintains the expressiveness afforded to models of the brain by deep neural networks.

In this paper we:

- introduce a new model of the hippocampal formation which learns the latent structure of an incoming stream of sensory stimuli analogous to a Helmholtz machine.
- describe a biologically plausible learning regime: Theta-oscillations gate information flow through multi-compartmental neurons which rapidly switches the system between "wake" and "sleep" phases. All plasticity is local.

- train our model on stimuli from a biologically relevant spatial exploration task and show it learns to path integrate by developing a ring attractor connectivity structure (comparable to theoretical predictions and empirical results in deep recurrent neural networks trained with gradient descent). Learning generalises: when the agent moves to a new environment, path integration capabilities recover without needing to relearn the path integration weights.

Our model of the hippocampal formation simultaneously (i) accounts for its role as a generative model of sensory stimuli, (ii) can learn to path integrate and (iii) can transfer structural knowledge between environments. The model, though here applied to the hippocampus, can be viewed as a step towards a general solution for how biological neural networks in many brain regions (for example visual cortex [10]) can learn generative models of the world.[1]

## 1.1 Related work

A recent generative model of the hippocampus, the Tolman-Eichenbaum Machine [23], proposed that the hippocampal formation be thought of as a hierarchical network performing latent state inference. Medial entorhinal cortex (MEC) sits atop the hierarchy and learns an abstract representation of space which is mapped to the hippocampus (HPC) where it is bound onto incoming sensory stimuli. Once trained the system can act in a generative fashion by updating the hidden representation with idiothetic action signals and then predicting the upcoming sensory experience. The drawback of this model, and others which share a similar philosophical approach [32, 24], is that it requires training via backpropagation through time (or equivalent end-to-end optimisation schemes, as in [24]) without clear biological correlates. Related hierarchical network architectures have also been studied in the context of reinforcement learning [33] and hippocampal associative memory [34].

Historically, hippocampal models of path integration have focused on continuous attractor networks (CANs) [25, 26, 27, 21] in entorhinal cortex. A bump of activity representing location is pushed around the CAN by speed and/or head-direction selective inputs, thus integrating self-motion. CANs have received substantial experimental support [28] but few studies adequately account for *how* this structure is learned by the brain in the first place. One exception exists outside the hippocampal literature: Vafidis et al. [35] built a model of path integration in the fly head-direction system which uses local learning rules. Here we go further by embedding our path integrator inside a hierarchical generative model. Doing so additionally relaxes the assumption (made by Vafidis et al. [35] and others [36]) that sensory inputs into the path integrator are predefined and fixed. Instead, by allowing all incoming and outgoing synapses to be learned from random initalisations, we achieve a more generalisable model capable of transferring structure between environments (see section 3.3).

Hippocampal theta oscillations have been linked to predictive sequence learning before [37, 38, 39, 40] where research has focused on the compressive effects of theta *sequences* and how these interplay with short timescale synaptic plasticity. Instead of compression, here we hypothesize the role of theta is to control the direction information flows through the hierarchical network.

Finally, a recent theoretical work by Bredenberg et al. [41] derived, starting from principles of Bayesian variational inference, a biologically plausible learning algorithm for approximate Bayesian inference of a hierarchical network model built from multi-compartmental neurons and trained with local learning rules using wake-sleep cycles. Here we build a similar network to theirs (i) extending it to a spatial exploration task and mapping the hidden layers onto those in the hippocampal formation, (ii) simplifying the learning rules and relaxing a discrete-time assumption – instead, opting for a temporally continuous formulation more applicable to biological tasks such as navigation – and (iii) adapting the hidden layer to allow idiothetic action signals to guide updates (aka. path integration). Their work provides a theoretical foundation for our own, helping to explaining *why* learning converges on accurate generative models.

## 2 A biologically plausible generative model trained with rapidly switching wake-sleep cycles and local learning rules

In sections 2 and 3 we give concise, intuitive descriptions of the model and experiments; expanded details can be found in the supplementary material.

---

[1] Code provided at `https://github.com/TomGeorge1234/HelmholtzHippocampus`

## 2.1 Basic model summary

We consider learning in an environment defined by a latent state, $z(t)$, which updates according to stochastic dynamics initially unknown to the network,

$$\frac{dz}{dt} = f_z(t). \tag{1}$$

These dynamics depends on the task; first we consider $z(t)$ to be a set of mutually independent random variables and later we consider the more realistic task of an agent moving on a 1D track.

The network recieves sensory input which is a function of the latent state into a sensory layer, $\mathbf{p}(t)$, and communicates this to a hidden layer (aka "internal state"), $\mathbf{g}(t)$. The network contains both an *inference* (aka. *recognition*) model which infers the hidden state from the sensory input (green arrows, Fig. 1a) and a *generative* model which updates the hidden state with recurrent synapses and maps this back to the sensory layer (blue arrows). As we will soon identify these processes with B̲asal and A̲pical dendritic compartments of pyramidal neurons we label activations sampled from the inference model with the subscript $B$ and those from the generative model with the subscript $A$. [2] In summary

$$\left.\begin{array}{l} \mathbf{p}_B(t+\delta t) = \bar{\mathbf{p}}(z(t)) \\ \mathbf{g}_B(t+\delta t) = \sigma_{g_B}(\mathbf{w}_{g_B}\mathbf{p}(t)) \end{array}\right\} \quad \text{Inference model} \tag{2}$$

$$\left.\begin{array}{l} \mathbf{g}_A(t+\delta t) = \sigma_{g_A}(\mathbf{w}_{g_A}\mathbf{g}(t)) \\ \mathbf{p}_A(t+\delta t) = \sigma_{p_A}(\mathbf{w}_{p_A}\mathbf{g}(t)) \end{array}\right\} \quad \text{Generative model.} \tag{3}$$

$\mathbf{w}_{g_B}, \mathbf{w}_{p_A}, \mathbf{w}_{g_B}$ are matrices of randomly initialised and plastic synaptic weights. $\bar{\mathbf{p}}$ maps the environmental latent into a vector of neural inputs. $\sigma$'s denote activation functions applied to the dendritic pre-activations – either the identity ($\sigma(x) = x$) or rectified tanh functions ($\sigma(x) = \max(0, tanh(x))$). A small amount of noise is added to the dendritic activations to simulate realistic biological learning.

We believe that the widely adopted convention of modelling neurons as single-compartment perceptrons is limiting. By considering, in a minimal extension, the distributed dendritic structure of real neurons we can tap into significant potential for explaining hippocampal learning. Theoretical [42, 43, 44, 45] and experimental [46, 47, 48] research into credit assignment in biological neurons has identified different roles for basal and apical dendrites: basal dendrites are thought to receive bottom-up drive from sensory inputs whereas apical dendrites receive top-down drive from higher layers in the sensory hierarchy [49]. Following this line of research — and matching an equivalent theoretical model of latent state inference described by [41] — we identify the inference process with synaptic inputs into a basal dendritic compartment of pyramidal neurons and the generative process with synaptic inputs into an apical dendritic compartment. In summary, each $\mathbf{p}$ and $\mathbf{g}$ neuron in our model has three compartments: a somatic compartment, a basal dendritic compartment and an apical dendritic compartment (Fig. 1b). Only the somatic activation is used for communication between layers (right hand side of Eqns. (2) and (3)) while dendritic compartment activations are variables affecting internal neuronal dynamics and learning as described below (Eqns. (4) and (6)).

## 2.2 Theta oscillations gate the direction of information flow through the network

The dynamics of the somatic activations $\mathbf{p}(t)$ and $\mathbf{g}(t)$ are as follows: the voltage in each soma is either equal to the voltage in the basal compartment *or* the voltage in the apical compartment depending on the phase of an underlying theta oscillation. This is achieved by a simple theta-gating mechanism (Fig. 1b):

$$\begin{array}{l} \mathbf{p}(t) = \theta(t)\mathbf{p}_B(t) + (1-\theta(t))\mathbf{p}_A(t) \\ \mathbf{g}(t) = \theta(t)\mathbf{g}_B(t) + (1-\theta(t))\mathbf{g}_A(t). \end{array} \tag{4}$$

where $\theta(t)$ is a 5 Hz global theta oscillation variable defined by the square wave function:

$$\theta(t) = \begin{cases} 1, & \text{if } t/T \mod 1 \leq 0.5 \\ 0, & \text{if } t/T \mod 1 > 0.5 \end{cases} \tag{5}$$

---

[2]These labellings conveniently match the notion that inferences are made from layers B̲elow in the sensory hierarchy (bottom-up) whereas generative predictions arrive from A̲bove (top-down).

for $T = 1/f_\theta$ and $f_\theta = 5$ Hz, matching the hippocampal theta frequency (5-10 Hz) [50]. According to this model theta-band oscillations in the hippocampal local field potential gate which dendritic compartment drives the soma. Experimental [47, 51, 52] and modelling work [53] gives provisional support for this assumption.

These local theta-dynamics have global consequences: the early phase ($\theta(t) = 1$) of each theta cycle can be thought of as a "wake" phase where information flows upwards through the network from the environment to the hidden layer, sampling the inference model. The latter phase ($\theta(t) = 0$) of each theta cycle is a "sleep" phase where information flows down from the hidden layer to the sensory units, sampling the generative model. These dynamics are displayed in Fig. 1.

## 2.3 Hebbian-style learning rules train synapses to minimise local prediction errors

In contrast to comparable models which are optimised end-to-end using backpropagation through time our model learns synaptic weights according to a local plasticity rule which is a simplified variant of a rule proposed by Urbanczik and Senn [43]. Incoming synaptic projections are continually adjusted in order to minimize the discrepancy between the somatic activation and the dendritic activation. The full learning rules are described in the supplement but simplified versions are given here:

$$\frac{d\mathbf{w}_{g_B}}{dt} \propto (\mathbf{g}(t) - \mathbf{g}_B(t))\mathbf{p}(t)^\mathsf{T}$$

$$\frac{d\mathbf{w}_{p_A}}{dt} \propto (\mathbf{p}(t) - \mathbf{p}_A(t))\mathbf{g}(t)^\mathsf{T}$$

$$\frac{d\mathbf{w}_{g_A}}{dt} \propto (\mathbf{g}(t) - \mathbf{g}_A(t))\mathbf{g}(t)^\mathsf{T} \tag{6}$$

Notably this learning rule is equivalent for *all* plastic synapses in the model: $\mathbf{p}$ to $\mathbf{g}$, $\mathbf{g}$ to $\mathbf{p}$ and the recurrent $\mathbf{g}$ to $\mathbf{g}$ synapses (see Fig. 1b). If a local prediction error is detected, for example the somatic activation is larger than the dendritic activation, then the synaptic strength of inputs into that dendritic compartment which are positive/negative are strengthed/weakened to reduce the error. This model can equivalently be viewed as a type of Hebbian learning – weight change is proportional to the correlation of pre- and post-synaptic activity (the first term) – regularised (by the second term) to prevent unbounded growth.

During the wake phase the weights of the generative model ($\mathbf{w}_{p_A}$ and $\mathbf{w}_{g_A}$) are trained and plasticity on the inference weights ($\mathbf{w}_{g_B}$) falls to zero. This occurs naturally because $\mathbf{p} = \mathbf{p}_B$ so there will be no basal prediction errors to correct. During sleep the reverse occurs; the weights of the inference model are trained and plasticity on the generative model falls to zero. Experimentally, apical activity is known to guide plasticity at basal synapses in CA1 [46]. This alternating, coordinated regime of sampling and learning (sample-inference-train-generative, then sample-generative-train-inference) is a hallmark of the wake-sleep algorithm. It fundamentally differs from the forward and backward sweeps of backpropagation since neurons remain provisionally active at all times so the process of learning minimally perturbs perception. Also, whereas backpropagation sends error signals down through the network to train synaptic weights, here only predictions are sent between layers and error signals are calculated locally at each dendrite.

As discussed in section 1, Bredenberg et al. [41] mathematically derive learning rules similar to these starting from a loss function closely related to the evidence lower bound (ELBO). As such our identification of early- and late-theta phases as "wake" and "sleep" cycles can be considered precise: from a Bayesian perspective our hippocampal model is minimising a modified ELBO loss (see supplement) thus learns to find approximately optimal inference and generative models accounting from the temporally varying stimulus stream it is presented.

## 2.4 Velocity inputs into the hidden layer

For path integration, the hidden state needs access to an idiothetic (internally generated) velocity signal. To satisfy this we endow the hidden layer, $\mathbf{g}$, with conjunctive velocity inputs, henceforth "conjunctive cells", as shown in Fig. 3a & b. Conjunctive cells are organised into two groups: $\mathbf{g}_{v_L}$ is responsible for leftward motion and $\mathbf{g}_{v_R}$ for rightward motion. Each conjunctive cell receives input from the hidden units and either the leftward ($v_L = \max(0, -\dot{x})$) or rightward ($v_R = \max(0, \dot{x})$) component of the velocity. For the results shown this connectivity is one-to-one $[\mathbf{w}_{g_{v_L}}]_{ij} = [\mathbf{w}_{g_{v_R}}]_{ij} = \delta_{ij}$ but

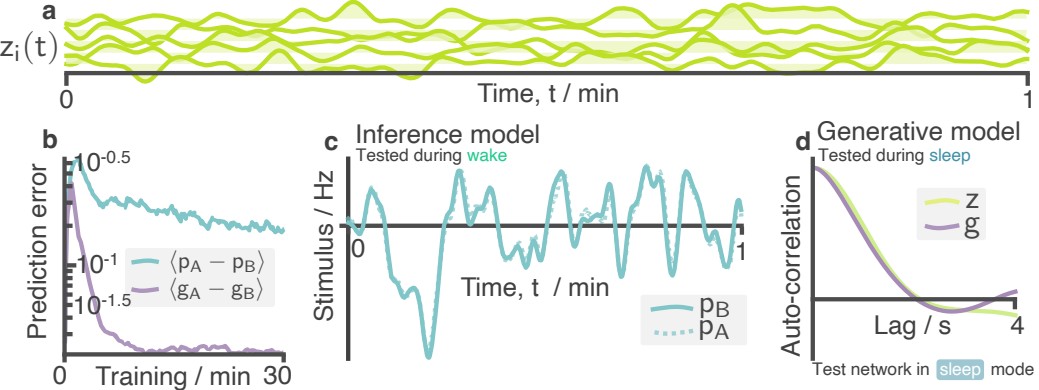

Figure 2: Learning in an environment of temporally varying latents. **a** In this artifical task the latent space comprises of $N_z = 5$ independent random variables with an autocorrelation decay timescale of 1 s. **b** Prediction errors (difference between apical and basal activations) in sensory and hidden layers reduce over training time. **c** Tested in wake mode ($\theta = 1$) after training, the ground truth stimulus matches apical prediction for all stimulus dimensions (one shown) implying the network is efficiently "autoencoding" the sensory inputs into and back out of the compressed hidden layer. **d** Tested in sleep mode ($\theta = 0$, no environmental inputs), generated data from the hidden units, $g$, have an autocorrelation curve which matches that of the true latents implying a statistically accurate generative model has been learned. More extensive samples from this model, before and after training, can be found in Fig. S1.

random connectivity works too, see supplement. Finally, conjunctive cells send return connections back to the apical dendritic compartment of the hidden units via a randomly initialised plastic synaptic weight matrix. This inputs are what drive the hidden units to path integrate.

This model takes inspiration from so-called conjunctive grid cells [54] found in the medial entorhinal cortex (MEC). These cells, though to be an integral component of the mammilian path integration system[27], are jointly tuned to head direction and location much like the conjunctive cells in our model. An important and novel aspect of our model is that synaptic weights between or into the hidden units are *learned*. This deviates from other models for example that by Burak and Fiete [27] (where all connectivity is predefined and fixed) or Vafidis et al. [35] and Widloski and Fiete [36] (where sensory inputs to the hidden units are pre-defined and fixed). This is not only more realistic but affords the model flexibility to translate path integration abilities between environments without having to relearn them, a form of transfer learning which we demonstrate in section 3.3.

## 3  Results

### 3.1  Validation on an artifical latent learning task

We begin by testing the basic model (i.e. without conjunctive inputs, Fig. 1a) on an artificial task. $N_z = 5$ latents, $z_i(t)$, are independently sampled from a smooth, random process with an autocorrelation timescale of 1 second (Fig. 2a). The sensory layer, $N_p = 50$, then receives a high-dimensional random linear mixture of the latents into the basal compartments:

$$\mathbf{p}_B(t) = \mathbf{A}\mathbf{z}(t), \tag{7}$$

where $\mathbf{A} \in \mathbb{R}^{50 \times 5}$ and $[\mathbf{A}]_{ij} \sim \mathcal{N}(0, \frac{1}{\sqrt{N_z}})$. The hidden layer, $\mathbf{g}(t)$, is matched in size to the latent process, $N_g = N_z = 5$, and all dendritic activation functions are linear. We train the model for 30 minutes of simulated time and track prediction errors, the difference between the basal and apical activations in the sensory and hidden layers, which reliably decreased throughout training (Fig. 2b). We then perform two tests designed to confirm whether the model has learnt accurate inference and generative models.

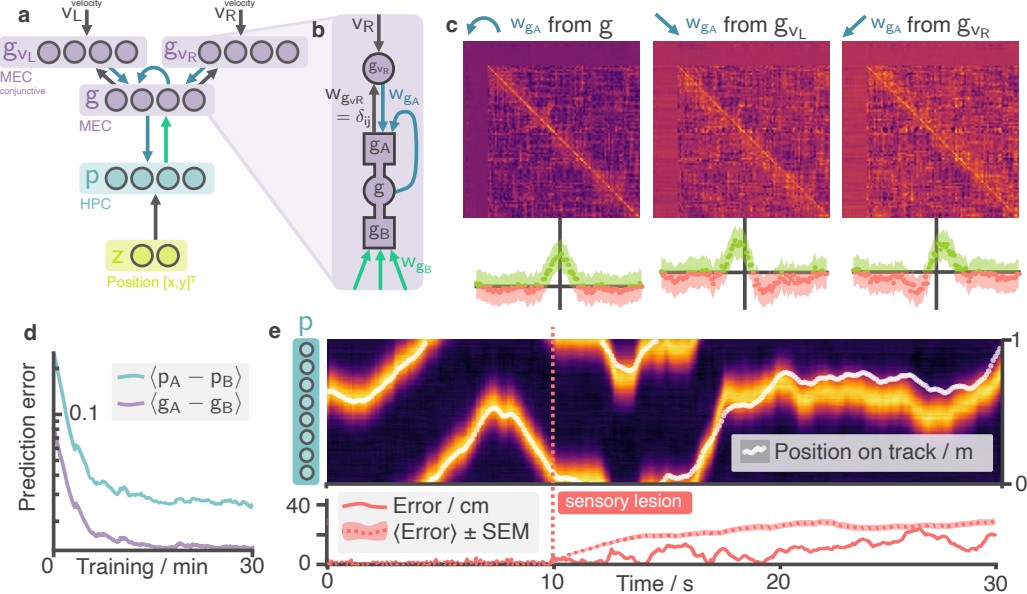

Figure 3: The hippocampal model learns to path integrate on a 1D track using a ring attractor. **a** Position selective (place cell) inputs drive basal dendrites of the sensory layer **p** (HPC). **b** Hidden units (MEC) are connected to two sets of "conjunctive cells" which each connect back to one of the hidden neurons (**g**) and either the leftward (for $\mathbf{g}_{v_L}$) or rightward (for $\mathbf{g}_{v_L}$) velocity of the agent allowing velocity information to enter the network. Synaptic strengths of the return connections from the conjunctive cells to the MEC hidden units, as well as those for the MEC recurrent connectivity (collective denoted $\mathbf{w}_{g_A}$), are randomly initialised and plastic. **c** After training, reordering the hidden units by the position of peak activity reveals a ring attractor in the synaptic weight matrices. Centre-surround recurrent connectivity stabilises an activity bump which is then "pushed" around the attractor manifold by asymmetric connections from the conjunctive cells, integrating velocity. Bands of zero weights show MEC neurons which have become perpetually inactive (aka "died"). The bottom panel displays the matrix row-averages, utilizing the circular symmetry of the environment to align rows before averaging. **d** Learning plateaus after 15 mins of simulated time. **e** Path integration ability is demonstrated in a lesion study: after 10 seconds in the normal oscillatory mode the network is placed into sleep mode (aka generative mode), lesioning the position-dependent sensory inputs. Despite this HPC continues to accurately encode position, evidence that the MEC ring attractor is path integrating the velocity inputs and sending predictions back to HPC. Lower panel shows the accumulated decoding error as well as the mean±SEM over 50 trials.

First, we set the dynamics of the model to "wake" mode ($\theta = 1$) and measure the basal and apical activations of one of the sensory neurons for 60 seconds. Close correspondence (Fig. 2c) confirms that the network accurately "autoencodes" the high-dimensional sensory inputs through the compressed hidden layer. Since all activation functions are linear this implies that $\mathbf{w}_{g_B}$ and $\mathbf{w}_{p_A}$ are pseudoinverses. Next, we place the network in "sleep" mode ($\theta = 0$) and allow the generative model to run freely. The autocorrelation of the generated hidden states ($\mathbf{g}(t|\theta = 0)$, displayed fully in the supplement) match that of the true environmental latents ($\mathbf{z}(t)$), Fig. 2d, implying the generative model has statistics closely matching those of the true underlying generative process.

## 3.2 Learnable path integration with a hidden ring attractor

Next we turn our attention to the hippocampal formation's role in spatial navigation, and our central result. The environment consists of an agent randomly moving around a 1 m 1D circular track (motion and cell data is generated using the RatInABox package [55]). The basal compartment of each HPC neuron is spatially tuned to a single different Gaussian input however non-Gaussian

randomly spatially tuned inputs work as well (see supplement Fig. S2b):

$$[\mathbf{p}_B(t)]_i = \exp\left[ -\frac{(x(t) - x_i)}{2\sigma^2} \right]. \tag{8}$$

$x(t)$ is the position of the agent and $\{x_i\}_{i=1}^{N_p}$ are the centres of the Gaussian inputs ($\sigma = 6$ cm), intended to simulate hippocampal place fields, evenly spaced at 1 cm intervals along the track. MEC (i.e. the hidden layer, $\mathbf{g}(t)$) is matched in size $N_g = N_p = 100$ with rectified tanh activation functions on both dendritic compartments ($\sigma_{g_B}(x) = \sigma_{g_A}(x) = \max(0, \tanh(x))$) and HPC (the sensory layer $\mathbf{p}(t)$) is linear ($\sigma_{p_A}(x) = x$). Two populations of conjunctive cells (Fig. 3a & b) feed into the apical compartments of the MEC recurrent units. Random initialisation of $\mathbf{w}_{g_B}$ means that MEC neurons start off with random non-Gaussian spatial tunings. $\mathbf{w}_{g_A}$ and $\mathbf{w}_{p_A}$ are also randomly initialised.

The network is trained for 30 minutes with learning plateauing after 15 (Fig. 3d). A lesion study, designed to test path integration, is then performed as follows: First, the network is run for 10 seconds normally (i.e. with theta-oscillating periods of wake and sleep). Since the simulated HPC neurons receive place-tuned inputs uniformly ordered along the track (i.e. $x_j > x_i \forall i, j > i$) an activity heatmap of HPC reveals a bump of activity accurately tracking agent's position (Fig. 3e, left). The network is then placed into a sleep phase ($\theta = 0$) for 20 seconds. This amounts to a full sensory lesion since top-down MEC inputs, not bottom-up place-tuned sensory inputs, drive HPC. Despite the full sensory lesion, hippocampal activity remains approximately unperturbed and the activity bump continues to accurately track position, slowly accumulating errors (Fig. 3e right). Since our HPC layer has no recurrent connectivity it cannot support this post-lesion activity on its own. Instead feed-forward drive from an MEC ring attractor, which we turn our attention to now, is responsible for maintaining the HPC code.

To find the ring attractor we must first reorder the MEC cells. We do this according to the position of the peak of their receptive fields (defined in the supplement). After reordering, the recurrent connectivity matrix can be seen to have acquired a centre-surround connectivity profile. Nearby MEC cells were, on average, strongly and positively recurrently connected to one another. Those far apart weakly inhibit one another (Fig. 3c, left; band of strong positive weights along diagonal flanked by weak negative weights). This profile matches that of a quasi-continuous ring attractor: local excitatory and long-range inhibitory connections stabilise a bump of activity on the attractor manifold in the absence of sensory input [56]. Weights from the conjunctive cells acquired asymmetric connectivity (Fig. 3c, middle & right) skewed towards the velocity direction for which they are selective. These asymmetric connections enable conjunctive cells to "push" the activity bump around the manifold, integrating velocity (see supplement for a visualisation of the MEC bump attractor). Theoretical work on ring attractors has demonstrated that for accurate path integration the asymmetric weights must be proportional to the derivative of the symmetric weights [56], approximately observed here. A noteworthy observation is that some MEC neurons become perpetually inactive; this is a consequence of the fact that *both* top-down and bottom-up synapses into the hidden layer are plastic and can fall to zero (Fig. 3c bands of zero-weights) satisfying a trivial $g_A = g_B = 0$ solution for minimising the prediction error. Despite this, not all MEC neurons die and the surviving subset are sufficient for path integration. In supplementary section 5.4.2 we discuss additional results showing when the network learns robust path integrate under a variety of plasticity, initialisation and noise manipulations.

Crucially, what sets this model apart from others [19, 20, 21, 22] is that the network is not optimized using a conventional path-integration objective and backpropagation. Instead, it has been demonstrated how path integration can naturally arise in a biologically constrained network subject to a much simpler (yet more broadly applicable) local objective, in cases where idiothetic velocity signals are available to the hidden layers.

### 3.3 Remapping: transfer of structural knowledge between environments

Finally, we demonstrate how our trained network can transfer structural knowledge – which here means the ring attractor and thereby path integration – between environments. We start by training the network as in section 3.2; the only diffence is that for simplicity we choose to fix $\mathbf{w}_{g_B} = \delta_{ij}$ giving rise to MEC representations which, like HPC, are unimodal (this constraint can be relaxed and, in the more general case, MEC units typically have multiple receptive fields, Fig S4d, reminiscent of grid cells). We then simulate a hippocampal "remapping" event by shuffling the sensory inputs to the HPC layer (Fig. 4a & b, top panel) and retraining the network for a further 30 minutes but this time

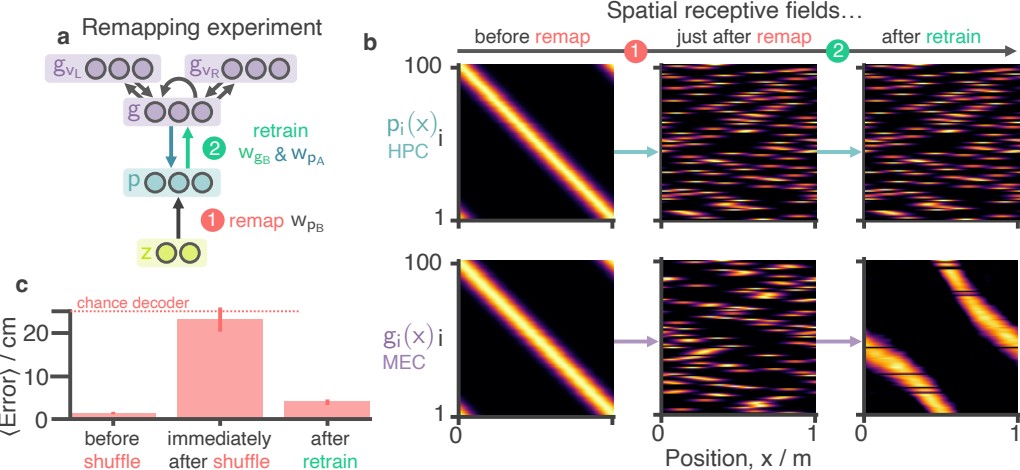

Figure 4: Remapping and transfer of structural knowledge between environments. **a** After training (as in Fig. 2) place cell inputs are shuffled to simulate a "remapping" event observed when an agent moves to a new environment. The agent then retrains for an additional 30 minutes: during this period internal MEC weights, and weights from the conjuctive cells to MEC are held fixed while MEC ↔ HPC weights remain plastic. **b** Recptive fields of the HPC and MEC neuronal populations at different stages in the experiment: Initially after remapping HPC and MEC inputs are randomised. MEC relearns rate maps as they were before remapping but with a constant phase shift. Note: neurons are ordered by the position of their peak activity on the track *before* remapping and this ordering is maintained in subsequent panels. **c** The error (± SEM over 50 trials) after 1 second of path integration is shown at different stages of the experiment. Although path integration is initially disrupted after remapping it recovers despite no relearning of the MEC synapses where the ring attractor is stored.

holding weights in the hidden layer, $\mathbf{w}_{g_A}$. Only the HPC ↔ MEC synapses ($\mathbf{w}_{g_B}$ & $\mathbf{w}_{p_A}$) remain plastic during retraining. Biologically this may be accounted for by the observation that cortical plasticity is substantially slower than hippocampal plasticity [57].

During biological remapping events place cells remap independently whereas grid cells remap *en masse* with entire modules shifting by the same constant phase [58]. This observation is reproduced in our model: after retraining MEC units regroup with receptive fields as they were before remapping but with a constant phase shift along the track. This re-emergence of structure occurs because the ring attractor seeds a bump of activity on the attractor manifold (during the "sleep" phases of retraining) onto which the shuffled HPC inputs then bind. Since nothing constrains *where* on the circularly symmetric attractor manifold this regrouping can initiate, only relative correlations, modulo a phase shift, are preserved.

Decoding error one second after a sensory lesion is tested just *before* remapping, just *after* remapping and after retraining (Fig. 4c). After the remapping path integration abilities temporarily disappear because the MEC ring attractor is still tuned to the old and invalid HPC receptive fields. After relearning – and despite *no adjustments to the MEC weights, $\mathbf{w}_{g_A}$, where the ring attractor is stored* – path integration recovers to almost the level before remapping. This differs substantially from other local models of path integration learning [35, 36] which don't consider plasticity on the ring attractor inputs. In these models, adaptation to a new environment necessarily requires complete relearning of the ring attractor. Instead our model exploits the basic fact that movement (path integration) in one environment is fundamentally the same as in another, one must simply learn a new mapping to/from the ring attractor, "translating" it to fit the new sensory stimuli.

## 4 Discussion

We propose that the hippocampal formation resembles a Helmholtz machine, simultaneously learning an inference and generative model of sensory stimuli. Like previous models [23] medial entorhinal

cortex (MEC) sits hierarchically above the hippocampus (HPC) to which it sends generative predictions. Our model differs in the learning rules and neural dynamics: local prediction errors are minimised between distinct dendritic compartments receiving bottom-up and top-down signals. Theta oscillations regulate internal neural dynamics, switching the network between wake and sleep phases. In a navigation task our MEC model forms a ring attractor capable of path integration. Despite simple learning rules and dynamics our model retains key cognitive capabilities of the hippocampal formation including the ability to transfer knowledge across different sensory environments.

Local learning rules are commonly recognised as essential in biologically plausible learning algorithms [43]. However, the importance of learning *scheduling* – how neural systems coordinate or multiplex distinct phases of forward and backward information flow – is often overlooked[59]. Neural oscillations such as theta, hypothesized to temporally coordinate communication between neuronal populations [60], likely play an underexplored role in this regard (neural "bursting" has also been pointed out as a potential solution to multiplexing [61]). One advantage of the wake-sleep algorithm, which this study suggests neural oscillations can support, compared to forward and backward sweeps is that, during convergence, the two phases become highly similar, allowing learning to proceed without affecting perception.

While our discussion has primarily focused on theta oscillations as a mechanism for learning, they have also been proposed as a mechanism for short-range future prediction via so-called "mind-travel"[62]. During the latter phase of each theta cycle (i.e. the sleep phase) gain amplified velocity signals might rapidly drive the MEC activity bump along the manifold allowing the agent to assess nearby upcoming locations. This complimentary proposition could neatly integrate into the framework proposed here and emphasizes the need for further investigation into the multifaceted functions of neural rhythms within the hippocampal/entorhinal system.

Beyond theta oscillations, both faster gamma cycles [63] and the slower physiological states of sleep and wake [64] have been associated with learning. Based on our model we suggest a tentative hypothesis that theta oscillations may be favored due to an optimality criterion; whilst faster oscillations could be a mechanism to prevent extreme drift during sleep that might disrupt learning their frequency might by upper bounded biophysically by the neural time constants associated with the biophysical processes supporting dendritic gating the soma. These ideas, their relevance to other brain regions involved in generative learning, 2D spatial dynamics, and offline memory consolidation/replay remain exciting questions for future theoretical and experimental investigation.

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
