# 5    Supplementary Material

## 5.1    Basic model description

Here we give a general decription of the model. Specifics for each experiment (i.e. learning rates, layer sizes, time constants etc.) are given in later sections.

### 5.1.1    Dendritic updates

Complete versions of the dendritic update rules (summarised in Eqns (2) & (3)) are given below. We assume dendrites recieve and integrate synaptic inputs according to the following dynamics:

$$
\left.\begin{array}{l}
\tau\frac{d\mathbf{p}_B(t)}{dt} = -\mathbf{p}_B(t) + \bar{\mathbf{p}}(z(t)) \\
\tau\frac{d\mathbf{g}_B(t)}{dt} = -\mathbf{g}_B(t) + \sigma_{g_B}(\mathbf{w}_{g_B}\mathbf{p}(t))
\end{array}\right\} \quad \text{Inference model} \tag{9}
$$

$$
\left.\begin{array}{l}
\tau\frac{d\mathbf{g}_A(t)}{dt} = -\mathbf{g}_A(t) + \sigma_{g_A}(\mathbf{w}_{g_A}\mathbf{g}(t)) \\
\tau\frac{d\mathbf{p}_A(t)}{dt} = -\mathbf{p}_A(t) + \sigma_{p_A}(\mathbf{w}_{p_A}\mathbf{g}(t))
\end{array}\right\} \quad \text{Generative model.} \tag{10}
$$

We discretise these dynamics in order to implement them computationally by making the common assumption that neural dynamics are fast ($\tau \approx 0$ ms) relative to the timescale of the synaptic inputs and so the compartments are always at equilibrium, recovering Eqns (2) & (3). This is valid in our regime where the environmental latent updates slowly compared to neural timescales. The notation we're using admits the possible presence of biases as well as the weights (though biases typically aren't used) by assuming a row of constant 1's could be added to the synaptic inputs effectively absorbing a bias into the weight matrix without loss of generality, for example $\mathbf{w}_{g_B}\mathbf{p}(t) \leftarrow \mathbf{w}_{g_B}\mathbf{p}(t) + b_{g_B}$.

### 5.1.2    Somatic updates

Somatic updates rules (Eqns (4) & (5)) and are repeated here for completeness:

$$
\begin{aligned}
\mathbf{p}(t) &= \theta(t)\mathbf{p}_B(t) + (1 - \theta(t))\mathbf{p}_A(t) \\
\mathbf{g}(t) &= \theta(t)\mathbf{g}_B(t) + (1 - \theta(t))\mathbf{g}_A(t).
\end{aligned} \tag{11}
$$

where $\theta(t)$ is a 5 Hz global theta oscillation variable defined by the square wave function:

$$
\theta(t) = \begin{cases} 1, & \text{if } t/T \mod 1 \leq 0.5 \\ 0, & \text{if } t/T \mod 1 > 0.5 \end{cases} \tag{12}
$$

### 5.1.3    Update ordering

For this hierarchical network of multicompartmental neurons we must specify the order in which we perform these discrete updates to the different layers and the different compartments within these layers. Strictly speaking, when the discretisation timestep $dt$ is small this ordering is arbitrary, but we include it here for completeness.

We update the layers from bottom to top: first we update the latent or "environment" and increment the global clock ($z(t+dt) \leftarrow z(t)$ & $t+dt \leftarrow t$). Next we update both dendritic compartments of the sensory layer ($\mathbf{p}_B(t+dt) \leftarrow \mathbf{p}_B(t)$ & $\mathbf{p}_A(t+dt) \leftarrow \mathbf{p}_A(t)$ noting that it makes no difference in which order these updates are done as they are independent. Then we update the somatic compartment of the sensory layer ($\mathbf{p}(t + dt) \leftarrow \mathbf{p}(t)$). Next we work upwards to the hidden layer ($\mathbf{g}_B(t + dt) \leftarrow \mathbf{g}_B(t)$ & $\mathbf{g}_A(t + dt) \leftarrow \mathbf{g}_A(t)$ followed by $\mathbf{g}(t + dt) \leftarrow \mathbf{g}(t)$) then, if present, the top-most "conjunctive cells" are updated. This gives the following dendritic update rules which are only slightly – and in the limit $dt \rightarrow 0$, irrelevantly – different from the simplified update rules given in the main text:

$$
\begin{aligned}
\mathbf{p}_B(t + dt) &= \bar{\mathbf{p}}(z(t + dt)) \\
\mathbf{p}_A(t + dt) &= \sigma_{p_A}(\mathbf{w}_{p_A}\mathbf{g}(t)) \\
\mathbf{g}_B(t + dt) &= \sigma_{g_B}(\mathbf{w}_{g_B}\mathbf{p}(t + dt)) \\
\mathbf{g}_A(t + dt) &= \sigma_{g_A}(\mathbf{w}_{g_A}\mathbf{g}(t))
\end{aligned} \tag{13}
$$

### 5.1.4 Learning rules

Learning rules are conceptually summarised by the equations given in the main text, Eqn (6). Here we give the *full* equations which include some adjustments to account for the presence of non-linear activation functions and temporal smoothing of the local prediction error learning signals. In our multilayer network all sets of learnable weights follow an equivalent learning rule. For this reason we choose give it here in its most general form: Consider the synaptic weight $w_{ij}$ connecting from the soma of presynaptic neuron $j$ with activation $f_j^{\text{pre}}$ to one of the dendritic compartments of a postsynaptic neuron $i$ with activation $f_{C,i}^{\text{post}} = \sigma(V_{C,i}^{\text{post}})$ (this could be the basal or apical compartment, $C \in \{A, B\}$). Weights are updated on each timestep by an amount:

$$\delta w_{ij}(t) = \eta \text{PI}_{ij}(t) \tag{14}$$

where $\text{PI}_{ij}$ is (following terminology used in Urbanczik and Senn [43]) the "plasticity induction" variable which is a low-pass filtered measure of the coincidence between the local prediction error and the synaptic input. The prediction error measures how far the activation of the dendritic compartment, $f_{C,i}^{\text{post}}$, is from the somatic activation $f_i^{\text{post}}$. In total, $\text{PI}_{ij}$ is defined by the following dynamics:

$$\tau_{\text{PI}} \frac{d\text{PI}_{ij}}{dt} = -\text{PI}_{ij} + \underbrace{[f_i^{\text{post}}(t) - f_{C,i}^{\text{post}}(t)]}_{\text{postsynaptic prediction error}} \cdot \sigma'(V_{C,i}^{\text{post}}(t)) \cdot \underbrace{f_j^{\text{pre}}(t)}_{\text{presynaptic input}} \tag{15}$$

If the prediction error and one of the presynaptic inputs are both consistently large (i.e. over a time period $\mathcal{O}(\tau_{\text{PI}})$) then the plasticity induction variable will therefore also be large and the weight connecting the pre- and postsynaptic neurons will be strengthed (thus decreasing future prediction errors). $\tau_{\text{PI}}$ is taken to be the same as used in Urbanczik and Senn [43], 100 ms. Note for fast filtering ($\tau_{\text{PI}} \to 0$ ms) and linear activation functions this reduces to the simplified formulae given in the main text, Eqn. (6).

### 5.1.5 Synaptic noise

We add synaptic noise to the dendritic activations. Each dendritic compartment maintains its own independent noise variable, $n(t)$, which is modelled as an Ornstein-Uhlenbeck process. The benefit of modelling neural noise with an Ornstein-Uhlenbeck process is that it is timestep size independent. The dynamics of the noise variable are given by:

$$n(t + dt) = n(t) + \frac{dt}{\tau}n(t) + \sqrt{\frac{2\sigma^2 dt}{\tau}}\xi(t) \tag{16}$$

where $\xi(t) \sim \mathcal{N}(0, 1)$ is a white noise process. These dynamics lead to a stationary distribution of $n(t)$ which is Gaussian with zero mean and variance $\sigma^2$. The decoherence timescale of the noise is $\tau$. We fix $\tau = 300$ ms and $\sigma = 0.01$ Hz in order that noise is relatively slow and weak. Noise is added at each timestep to the activation of the dendrites, e.g. $\mathbf{p}_B(t) \to \mathbf{p}_B(t) + \mathbf{n}_B(t)$ where $\mathbf{n}_B(t)$.

### 5.1.6 Measuring the prediction error

Figs. 2b & 3d show the prediction errors of the network layers decreasing throughout training. Here we define how these errors. A consequence of our learning rule is that during wake, the apical dendrites adjust to try minimise the discrepancy between the apical activation and the soma (which, during wake, is equal to by the basal activation). During the sleep phase a short time later the basal dendrites adjust to try minimise the discrepancy between the basal activation and the soma (which, during sleep, is equal to apical activation). If learning is successful we would expect the apical and basal activations to converge, thus we use the following measures of the prediction error to track training performance in both layers of the network:

$$\mathcal{E}_p(t) = \frac{1}{N_p} \sum_i |[\mathbf{p}_B(t)]_i - [\mathbf{p}_A(t)]_i|$$

$$\mathcal{E}_g(t) = \frac{1}{N_g} \sum_i |[\mathbf{g}_B(t)]_i - [\mathbf{g}_A(t)]_i|. \tag{17}$$

These are then smoothed with a decaying exponential kernel of timescale 60 seconds to remove some of the nosie and better display the learning signal.

## 5.2 Relationship to online Bayesian Inference

Bredenberg et al. [41] derived local synaptic learning rules for a similar hierarchical network performing online latent inference starting from a loss function closely related to the evidence lower bound (ELBO) of variational inference. Here we will not repeat their derivation, instead we intend to highlight their starting point, the most important assumptions they made and the learning rules they derived, finally pointing out how ours differ. The point is to demonstrate that the learning rules we propose are not arbitrary but can actually be derived from a more principled approach to online inference.

Bredenberg et al. [41] consider a network recieving input from a latent variable $z$. The network has two layers, $\mathbf{p}_t$ and $\mathbf{g}_t$. [3] The network is trained to perform online inference over a sequence of observations from the environment, $z_{0:T}$. To do this they start from the loss function

$$\mathcal{L} = \mathbb{E}_{\theta,\mathbf{z}}\big[D_{KL}(\tilde{q}_w \parallel \tilde{p}_w)\big] \tag{18}$$

where $\tilde{q}_w$ and $\tilde{p}_w$ are the following probability distributions over the layer variables $\mathbf{p}_t$ and $\mathbf{g}_t$:

$$\tilde{q}_w = \prod_{t=0}^{T} \left(q(\mathbf{g}_t|\mathbf{p}_t; w_{\text{inf}})p(\mathbf{p}_t|z_t)\right)^{\theta_t} p_m(\mathbf{g}_t, \mathbf{p}_t|\mathbf{p}_{t-1}, \theta_t; w_{\text{gen}})^{1-\theta_t}, \tag{19}$$

$$\tilde{p}_w = \prod_{t=0}^{T} \underbrace{\left(p(\mathbf{g}_t|\mathbf{p}_t; w_{\text{inf}})p(\mathbf{p}_t|z_t)\right)^{1-\theta_t}}_{\text{inference model}} \underbrace{p_m(\mathbf{g}_t, \mathbf{p}_t|\mathbf{p}_{t-1}, \theta_t; w_{\text{gen}})^{\theta_t}}_{\text{generative model}} \tag{20}$$

and $\theta_t \in \{0, 1\}$ is a binary variable (in their analysis they fix this to oscillate in fixed symmetric phases, e.g. 000111000111...). The two probability distributions, $\tilde{q}_w$ & $\tilde{p}_w$, which this loss function attempts to make similar to one another, can be interpreted as the probabilites over the layer variables $\mathbf{p}_t$ and $\mathbf{g}_t$ in two noisy neural networks [4] connected as we drew in Fig. 1a: the first network alternates between phases of inference, where information flows bottom up from the latents $z$ to the hidden layer $\mathbf{g}$, and generation, the opposite (inference-generation-inference-generation...), the second network alternates in exact counterphase (generation-inference-generation-inference...). This loss is a generalisation of the widely used evidence lower bound (ELBO) which corresponds to the case where $\theta_t = 1$ for all $t$. ELBO loss functions seek to make the inference and generative distributions over sensory and hidden variables similar. We will not delve further into the justifications for these types of loss functions other than to state that they are widely used[1].

One of the key conceptual steps taken by Bredenberg et al. [41] (and now us) is to note that processes of performing inference and generation can locally occur simultaneously as long as they are recieved into distinct dendritic compartments. Which dendrite then gates into the soma (i.e. Eqn 4) then dictates the global state (wake or sleep) of the network. It also means, as they show, that the loss can be approximately optimized using local learning rules by comparing the dendritic compartment activation to that of the soma. The learning rules they derive, again translated into our notation, are as follows (note for simplicity we assume all activations are linear since non-linearities add only one additional multiplicative term into their update equations, see equations (14), (15) and (16) in [41]):

$$\frac{d\mathbf{w}_{g_B}}{dt} \propto (1 - \theta_t)(\mathbf{g}_t - \mathbf{g}_{B,t})\mathbf{p}_t^\mathsf{T} \tag{21}$$

$$\frac{d\mathbf{w}_{p_A}}{dt} \propto \theta_t(\mathbf{p}_t - \mathbf{p}_{A,t})\mathbf{g}_t^\mathsf{T} \tag{22}$$

$$\frac{d\mathbf{w}_{g_A}}{dt} \propto \theta_t(1 - k_t)(\mathbf{g}_t - \mathbf{g}_{A,t-1})\mathbf{g}_{t-1}^\mathsf{T} \tag{23}$$

where $k_t = (1 - \delta(\theta_t - \theta_{t-1}))\theta_t$ is a term which is 1 if and only if $\theta_t = 1$ and $\theta_{t-1} = 0$ therefore it briefly turns off learning upon switching from sleep to wake.

Reader may like to compare these learning rules to our own as given in the main text Eqns (6). Our learning rules differ from theirs in the following way:

---

[3] For convenience we have translated their variables into our notation ($\mathbf{g} \leftrightarrow \mathbf{r}, \mathbf{p} \leftrightarrow \mathbf{s}, \theta \leftrightarrow \lambda, \mathbf{w} \leftrightarrow \theta$) so it is easier to compare.

[4] Note there isn't *actually* two networks being trained. Instead they use a mathematical trick, deriving from the symmetry in the alternating phase of the theta cycle, to do away with the need to sample from both networks meaning they can deriving local learning rules which can train a single network, e.g. $\tilde{q}_m$, on its own. This single network, like ours, contains both inference and generative models, represented by the two terms in equation (19)

- We relax their discrete time assumption, opting for a continuous time formulation ($\mathbf{p}_t \to \mathbf{p}(t)$ etc.).

- We note that the terms in the equations proportional to $\theta_t$ or $1 - \theta_t$ which actively turn on or off learning depending on whether $\theta_t = 0$ or $1$ are unnecessary since the prediction error term natural falls to zero anyway. For example, in Eqn. (22) when $\theta_t = 0$ the network is in sleep and so $\mathbf{p}_t = \mathbf{p}_{A,t}$. In this case the prediction error is zero by definition and learning ceases even without the preceeding $\theta_t$ term.

- We disregard the $1 - k_t$ term. Empirically this does not seem to damage our model and theoretically its impact should only be small in our continuous time formulation where the network is only switching from sleep to wake for a negligible proportion of the time.

- Upon provisional theoretical and experimental justification we liken $\theta$ the theta component of the hippocampal local field potential and set it to 5 Hz.

Ultimately these changes are surface level. Our learning rules can – and should – be understood as a close approximation to those derived by Bredenberg et al. [41]. Consequently it is appropriate to consider our hippocampal model as learning to perform approximately optimal online Bayesian inference.

## 5.3 Experiment 1: An artifical latent learning task

$N_z = 5$ independent, autocorrelated, random latent variables are sampled from a Gaussian process with a squared exponential covariance function of width 1 second, samples of these are shown in Fig. 2a and Fig. S2. The sensory layer is large ($N_p = 50$) relative to the compressed hidden layer ($N_g = N_z = 5$) and recieves a random mixture of the latents into the basal compartments as described in the text. All activation functions are linear, no layers have biases, all learning rates are set to $\eta = 0.01$, and the discretisation timestep was $dt = 25$ ms. Weights are initialised randomly $[\mathbf{w}_{\mathbf{g}_B}]_{ij} \sim \mathcal{N}(0, 1/\sqrt{N_p})$, $[\mathbf{w}_{\mathbf{p}_A}]_{ij} \sim \mathcal{N}(0, 1/\sqrt{N_g})$, $[\mathbf{w}_{\mathbf{g}_A}]_{ij} \sim \mathcal{N}(0, 0.1/\sqrt{N_g})$ where the smaller initialisation on the recurrent weights, $\mathbf{w}_{g_A}$, was chosen to prevent unstable dynamics.

Before learning – since weights are initialised randomly – basal and apical voltages in the sensory layer are unmatched when tested for a period in wake mode (Fig. S1a). When tested for a period in sleep mode, the small initialisation of the recurrent weights means the hidden layer cannot sustain activity (Fig. S1b, top) which decays and decorrelates rapidly in contrast to the true latents (Fig. S1c). Compare this to after learning where, during wake, basal and apical voltages in the sensory layer are closely matched implying accurate autoencoding through the compressed hidden layer. During sleep, the hidden layer generates sustained activity statistically similar to the true latents (they do not match because during sleep the true latents are not driving the network, even during wake we would only expect our network to represent the true latents in its latent space up to a linear rotation), i.e. its is functioning as a generative model. Note the only source of randomness driving stochasticity and activity in the network is the noise in the dendritic updates themselves.

## 5.4 Experiment 2: Learnable path integration with a hidden ring attractor

An agent randomly moves around a 1 m 1D circular track. The trajectory, $x(t)$, is sampled using the RatInABox[55] simulation package. This means that velocity is model as an Ornstein-Uhlenbeck process (see Eqn. (16)) with a decoherence timescale of $\tau = 0.7$ seconds and a standard deviation of $\sigma = 0.5 \text{ ms}^{-1}$. There are $N_p = N_g = 100$ neurons in both layers. The HPC dendritic activation function is linear ($\sigma_{p_A}(x) = x$) whilst both MEC dendritic compartments have rectified tanh activation functions ($\sigma_{g_B}(x) = \sigma_{g_A}(x) = \max(0, \tanh(x))$). Note the choice of activation function means MEC neurons have firing rate $\mathcal{O}(1 \text{ Hz})$. All learning rates are set to $\eta = 0.01$, the discretisation timestep was $dt = 25$ ms and only $\mathbf{p}_A$ & $\mathbf{g}_B$ have learnable biases.

We model $N_i = N_p = 100$ inputs which are tuned to the position of the agent according to the following Gaussian tuning curves (these roughly model place cells):

$$[\phi(t)]_i = \exp\left[ -\frac{(x(t) - x_i)}{2\sigma^2} \right]. \tag{24}$$

where $x_i$ are centres of the Gaussians evenly spaced along the track. These then linearly drive the basal dendritic compartments of the sensory neurons:

$$\mathbf{p}_B(t) = \mathbf{B}\phi(x(t)) \tag{25}$$

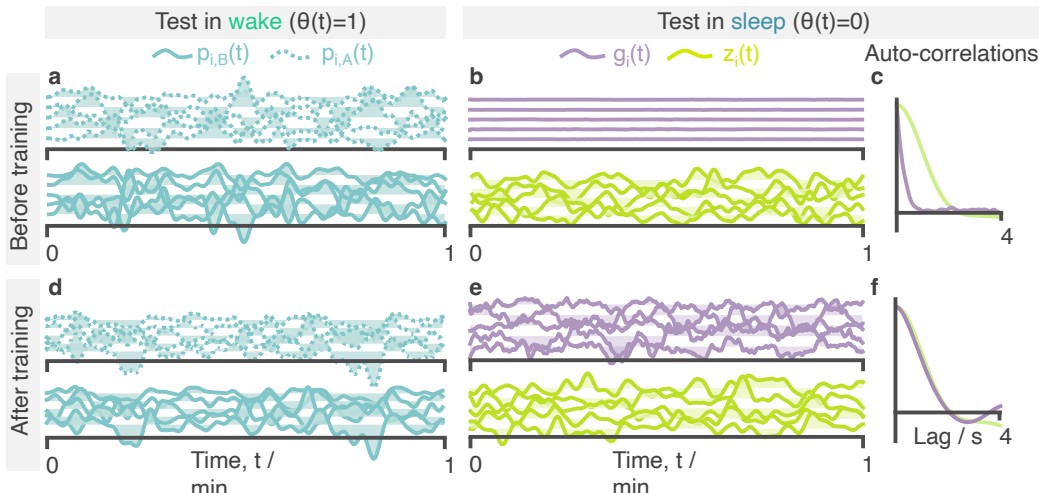

Figure S1: Extended results from the artificial latent learning task. **a** Basal and apical voltages in the sensory layer before learning during a one minute sample in wake mode. **b** Samples of activity in the hidden layer and true latents before training during a one minute sample in sleep mode. **c** Autocorrelations, averaged over the units, for activity in panel b. **d,e** & **f** As in a, b & c but after training.

where, in the results shown in the main paper, $\mathbf{B}_{ij} = \delta_{ij}$ is the identity matrix such that each sensory neuron inherits a unimodel-tuning curve from one and only one of the inputs, i.e. what was stated in Eqn. (8). We show in the supplementary figure S2 that this choice is not particularly critical and the network can learn to perform path integration with random sensory drive ($[\mathbf{B}]_{ij} \sim \mathcal{N}(0, 1/\sqrt{N_p})$).

Velocity inputs are connected as follows: two neurons encode the rectified leftward and rightward velocity of the agent, normalised by the standard deviation $\sigma$. Note, this means they have firing rates $\mathcal{O}(1 \text{ Hz})$.

$$v_L(t) = \max(0, -\dot{x}(t)/\sigma)$$
$$v_R(t) = \max(0, \dot{x}(t)/\sigma) \tag{26}$$

Two sets of conjunctive cells ($N_g = 100$ in each set) sum inputs from the left and right velocity neurons and the hidden units as follows:

$$[\mathbf{g}_{v_L}(t)]_i = \sigma_{gv}\left(v_L(t) - v_R(t) + \sum_j [\mathbf{w}^{g_{vL}}]_{ij} [\mathbf{g}(t)]_j\right)$$

$$[\mathbf{g}_{v_R}(t)]_i = \sigma_{gv}\left(v_R(t) - v_L(t) + \sum_j [\mathbf{w}^{g_{vR}}]_{ij} [\mathbf{g}(t)]_j\right) \tag{27}$$

where $\sigma_{gv}(x) = \max(0, x - 1)$ is a ReLU function thresholded at $x = 1$. In the main paper we set $[\mathbf{w}^{g_{vL}}]_{ij} = [\mathbf{w}^{g_{vR}}]_{ij} = \delta_{ij}$ so each conjunctive cell is connected to one and only one hidden unit (something we relax in Fig. S2c). The consequence of this connectivity is that a $\mathbf{g}_{v_L}$ neuron is above threshold (and therefore active) if and only if the agent is moving to the left *and* the hidden unit it is connected to is active. Rightward motion silences $\mathbf{g}_{v_L}$ neurons. Similarly, a $\mathbf{g}_{v_R}$ neurons is active if and only if the agent is moving to the right and the hidden unit it is connected to is active. This conjunctive, logic-AND-gate-like tuning to both MEC and velocity is why these neurons are called "conjunctive" cells.

To order the MEC neurons after learning, and thus reveal the ring attractor, we calculate their receptive fields as a function of agent position, $\mathbf{g}(x)$, as though the network is in inference mode (so top-down recurrent connections and drive from the conjunctive cells do not play a role). Then we permute the ordering $i' \leftarrow i$ such that the maxima of the receptive feilds move from left to right along the track as the neuron count increases, $\arg\max_x [\mathbf{g}(x)]_{j'} > \arg\max_x [\mathbf{g}(x)]_{i'} \forall i', j' > i'$. The effect of this ordering procedure is shown in Fig. S2, panel a (left hand side, top two panels).

Fig S2a repeats the same path integration test as was shown in the main text Fig. 3 except now we additionally visualise the receptive fields of HPC and MEC (after learning) and show timeseries of both HPC and MEC neurons during the test. Once MEC neurons are reordered by their maxima the ring attractor activity bump can be seen moving up at down the manifold of neurons, even after the sensory lesion. Note again how some MEC neurons have "died" and do not engage in the ring attractor dynamics, forcing the ring attractor manifold to live on the remain subset of MEC neurons.

### 5.4.1 Position decoding

To quantify the performance of path integration we train a decoder to estimate agent position directly from the HPC population vector. The decoder is trained on positon and activity data from the final 10 minutes of training, after learning had plateaued. The decoder we use is a Gaussian process regressor with a squared exponential kernel, the length scale of which is optimised during fitting. The decoder works well as can be seen in the path integration plots where, before the sensory lesion, the decoded position correctly and accurately tracks the true position.

### 5.4.2 Robustness of path integration to weight initialisations, plasticity lesions and noise

Since a central claim of our work is that the network can learn, *from random initialisations*, the correct connectivity required to perform path integration, it is important to question where and why weights in our model are not randomly initialised and plastic.

**Sensory weights**  The weights from the Gaussian tuned inputs to the HPC sensory neurons, $\mathbf{B}$ in Eqn. (25), must be non-plastic to prevent the network from rapidly converging on a trivial solution where all input weights fall to zero killing all activity in the network and trivially minimising the local predition errors. They do not, however, need to be the identity function as we chose. Fig. S2 panel b repeats the standard path integration experiment but with a network where $[\mathbf{B}]_{ij} \sim \mathcal{N}(0, 1/\sqrt{N_p})$, path integration is still learned without any problem. Ultimately this is not particularly surprising since the mapping from the spatially-tuned sensory inputs, $\phi$, to the ring attractor in the orginal formulation was already mixed once by the randomly initialised weights from HPC to MEC ($\mathbf{w}_{g_B}$). This just adds one additional layer of mixing.

**MEC to conjunctive cells**  We show in Fig. S2 panel c, that path integration is still learned even when the MEC to conjunctive cell weights are initialised randomly, $[\mathbf{w}_{g_{vL}}]_{ij} \sim \mathcal{N}(0, 1/\sqrt{N_g})$, $[\mathbf{w}_{g_{vR}}]_{ij} \sim \mathcal{N}(0, 1/\sqrt{N_g})$. We leave it to future work to investigate this result more thoroughly but comment that it is a notable relaxation on assumptions made in previous models [35, 27] that fine-tuned connectivity from MEC to the conjunctive cells is assumed a priori for path integration (connectivity which would presumably have to be genetically encoded, which seems unlikely). We suspect part of the reason our path integration is robust with respect to the setting of these weights is down to the ability for MEC to construct its own inputs from HPC. This might means the exact form of the activity bump inside the ring attractor can be tailored to fit the specific connectivity to the conjunctive cells – which is perhaps randomly determined during development – in a particular network.

**Plasticity lesions**  Path integration, as explored in section 3.2, requires fine tuning the recurrent weights in the hidden layer ($w_{g_A}$) and consequently fails when this plasticity is turned off (Fig. S3a). Intriguingly however, we find that path integration does not *strictly* require plasticity between HPC and MEC (as shown in Fig. S3b, echoing results in [35]). However, when such plasticity is removed, the apical input to HPC coming from MEC is unmatched to the sensory input HPC recieves from the environment. As such, any downstream system reading out position from the HPC code would only be able to do so during sleep or wake and not both. This is somewhat restrictive for a system hoping to use the hippocampal formation for online inference and planning. Hence, a primary role of interlayer plasticity between HPC and MEC in our model is to "translate" the environment-agnostic MEC code into the the environment-specific HPC code. This idea is discussed further in section 3.3.

### 5.5 Experiment 3 details: Remapping

To investigate remapping we first train our network to path integrate as described in the main paper. The only difference is that we fix the weights from HPC to MEC to the identity matrix ($[\mathbf{w}_{g_B}]_{ij} = \delta_{ij}$

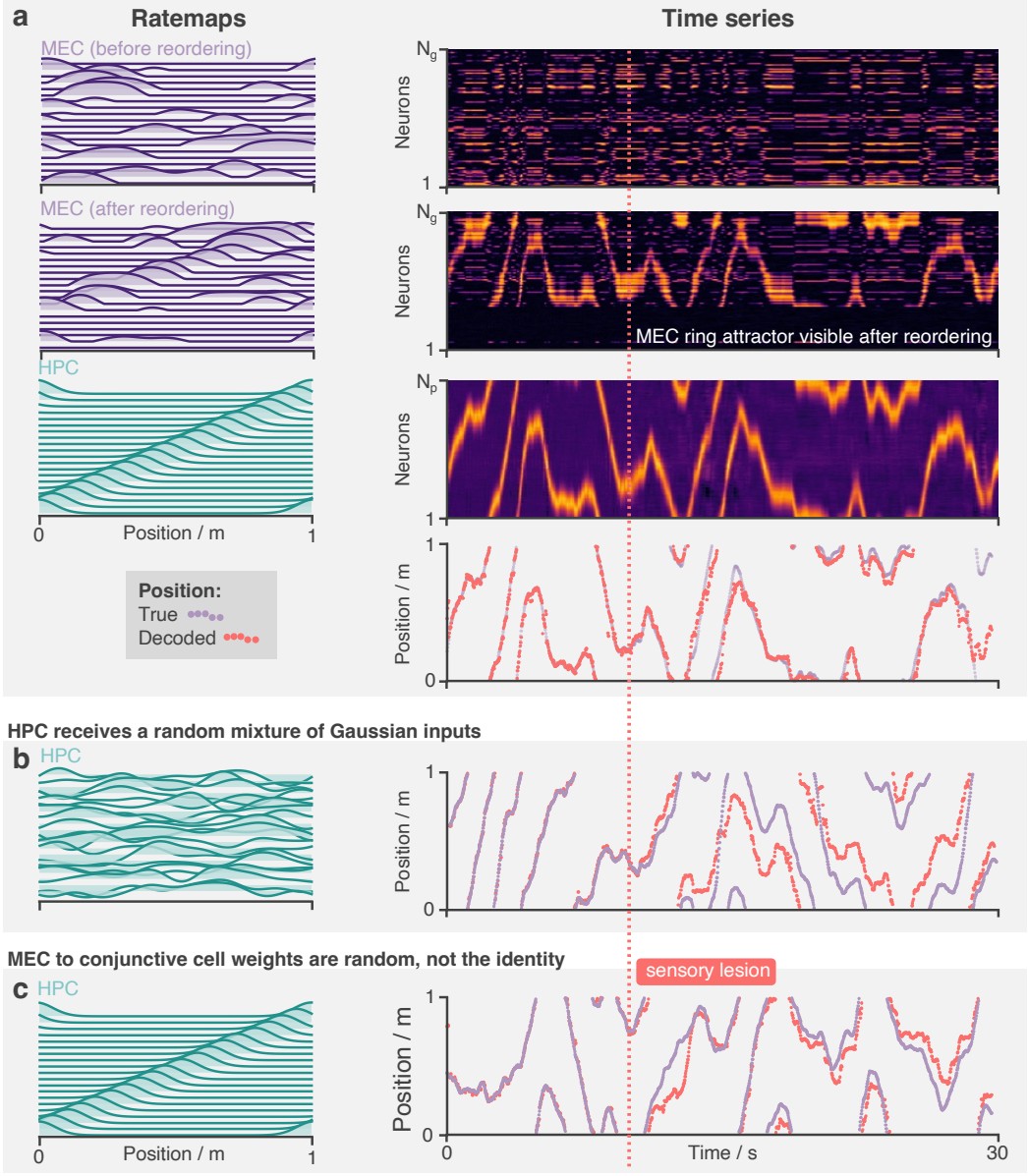

Figure S2: Path integration is performed by a ring attractor in MEC revealed once the neurons are reordered by receptive field peak position. The network learns to path integrate robustly, regardless of the choice of random initialisations. **a** The same path integration test as in the main text is performed here: The top three rows show receptive fields (left) and timeseries activity (right) for the MEC (top two) and HPC layers (third) layers. MEC receptive fields and activity at first appears random. It is only after reordering the neurons by the peak position of their receptive fields that we see the ring attractor manifold. The bottom row shows the decoded position (red) and the true position (purple), demonstrating accurate path integration. **b** Like panel a except, instead of unimodel Gaussian inputs, the HPC neurons recieve a random-sum-of-Gaussian inputs. Nonetheless the network still learns to path integrate (right). **c** Like panel a – with HPC neurons returned to their original Gaussian receptive fields – except in this experiment the hidden units (MEC, **g**) are connected to the conjunctive cells randomly, not one-to-one. The network still learns to path integrate.

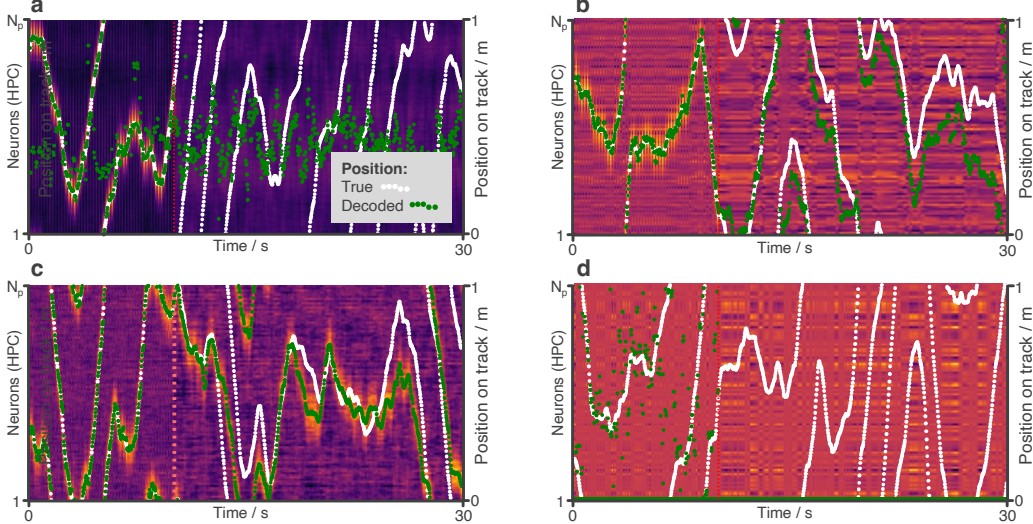

Figure S3: Network response to removal of plasticity and additional noise. The standard path integration experiment is perform and hippocampal activity (as well as true and decoded position) is shown in four modified conditions. **a** Plasticity on the recurrent synapses ($w_{g_A}$) is turned off and the network no longer learns to path integrate. **b** Plasticity on all weights between HPC and MEC ($w_{g_B}$ & $w_{p_A}$) is turned off. The network still learns to path integrate but inputs to HPC from MEC are not matched to those from the sensory input. **c** Synaptic noise on all synapses is increased by a factor of 10. The bump attractor is now noisier than Fig. 3e but path integration is still accurate. **d** Synaptic noise on all synapses is increased by a factor of 100 at which point learning fails.

and $\eta = 0$ on these weights) during this phase of training, this results in MEC neurons with receptive fields equal to those of the HPC neurons (except also passed through a rectified-tanh activation function), Fig 4b left column.

In the second phase we begin by randomly permuting the centres of the Gaussian sensory inputs in Eqn. (24). This "sensory shuffle" simulates the sort of hippocampal remapping event which typically occurs when an agent enters into a new environment. The activations of all neuronal layers are reset to zero. A second phase of learning then begins, this time only the weights from HPC to MEC ($\mathbf{w}_{g_B}$) and from MEC to HPC ($\mathbf{w}_{p_A}$) are plastic ($\eta = 0.01$) while the recurrent weights within MEC and the weights from the conjutive cells to MEC (collectively, $\mathbf{w}_{g_A}$) are frozen ($\eta = 0$).

We found that MEC neurons regroup after the shuffle, reestablishing the pairwise correlational structure they had before remapping with, perhaps, a phase shift (Fig. 4b). Once the ring attractor manifold has reappeared in this way the ability to path integrate returns (Fig. 4c). We find these results are clearest when $\mathbf{w}_{g_B}$ was fixed to the identity during the initial learning phase as desribed above. Although we don't investigate this finding thoroughly we suspect it is because the network has an easier time learning the ring attractor since the MEC inputs are already unimodal. With the identity mapping, a tidy activity bump already on the MEC cells before the rest of the ring attractor connectivity is learned, providing a good starting point. Note this matches the standard set up for studies of path integration in, for example, Vafidis et al. [35]). This, perhaps, leads to a ring attractor which is more deeply embedded into the MEC recurrent connectivity structure and which can therefore more easily reestablish itself after a remapping. Nonetheless we discover that MEC is able to relearn a significant portion of the bump attractor structure during the second phase of learning even when this was not the case and $\mathbf{w}_{g_B}$ was randomly intialised ($\mathbf{w}_{g_B} \sim \mathcal{N}(0, 1/N_p)$) and plastic during the initial learning, this is shown in Fig. S4. Note how, in contrast to the receptive field shown in Fig. 4b, the MEC neurons are now multimodal and additional bands of correlational structure (in addition to a global phase shift) appear after relearning. We leave it to future work to investigate this further.

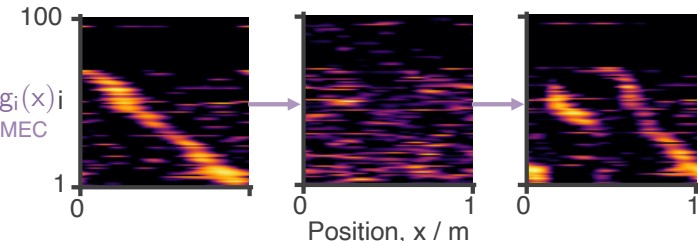

Figure S4: Regrouping of the MEC neurons after sensory remapping but relaxing the constraint that HPC to MEC weights are fixed to the identity matrix during initial learning. This results in MEC neurons with multimodal receptive fields and more complex regrouping dynamics after remapping.