# OpenReview forum: "A generative model of the hippocampal formation trained with theta driven local learning rules"
_NeurIPS.cc/2023/Conference — NeurIPS 2023 poster_

### Official Review · Reviewer_Lhyx · 2023-06-12

**Soundness:** 3 good
**Presentation:** 4 excellent
**Contribution:** 3 good
**Rating:** 7
**Confidence:** 4

**Summary:**

The paper is an application of the learning scheme derived in Bredenberg et al. 2021 to representation learning and path integration in the MEC and HPC. By adapting the aforementioned scheme to continuous time, and by relating the proposed oscillatory 'gating' signal to theta oscillations that have been observed to gate dynamics in HPC and MEC, the authors propose to model representation learning in these two brain areas as a form of rapidly alternating 'wake-sleep' learning where brief periods of generative prediction dynamics alternate with more standard position-driven dynamics and deviations between path-integrated predictions and sensory stimulus-driven inputs are used to drive learning at both apical and basal synapses. Subsequently they demonstrate that their MEC model is able to capture two interesting features, learned exclusively through local synaptic plasticity rules:

1. The MEC is able to learn a ring attractor capable of path integration based only on motion cues that is able to accurately predict subsequent dynamics.
2. This attractor can easily generalize to a new environment without modifications to its recurrent synaptic weights.

**Strengths:**

While previous models have successfully captured the features I previously mentioned through various optimization schemes, they have not shown how such learning could occur through local, biophysically-motivated synaptic plasticity which can be cleanly related to any representation learning objective function (in this case, the authors relate their learning to a variant of the ELBO objective). This is the principle success of the paper, and in my mind could prove to be a very important model for learning orchestrated between the MEC and HPC.

The authors also relax several constraints on the learning derived from Bredenberg et al. 2021:
1. They generalize learning dynamics to continuous time.
2. They include self-motion information in the 'generative' part of learning dynamics, which allows for path integration.
3. They relax constraints on gating of plasticity that are tied to theta cycles, simplifying learning and requiring fewer assumed variables that are locally available to synapses.

Furthermore, the paper is clearly written and logically presented.

**Weaknesses:**

To me, the principal weaknesses of the paper come from additional analyses that the authors could have done, but did not do. I will list several below:

1. The authors do not test their learning algorithm in more complex environments. Simple two-dimensional environments such as boxes or circles could be used to test whether or not the learning algorithm is sufficient to develop grid-like cells in MEC. More complex image-based environments could be used to test the scalability of the proposed learning algorithm, and could potentially test whether or not the proposed learning algorithm is also sufficient to develop place cell responses (rather than having them hand-provided for the system). Given that algorithms similar to Wake-Sleep do not scale as well to high-dimensional problems as backpropagation and that the brain is presumably able to learn place cell information indirectly from multimodal sensory input (vision, audio, vestibular inputs, etc.), this is a valid concern.

2. The authors do not discuss testable predictions for their learning algorithm--I can identify several, which I would like to see discussed in more detail. In particular, though the authors suggest that after learning has converged the 'generative' and 'inference' phases of learning 'become indistinguishable,' this is both untested and unlikely to be true for nondeterministic systems and agents operating in partially observable environments with stochastic transitions. In particular, conditioned on the system being in the generative phase, stochasticity in the neural network and uncertainty in the environment should cause the precision of the path integration estimate to decrease throughout time (as the system advances deeper into the theta phase). In fact, the authors' results (Fig. 3e) appear to show exactly this phenomenon. In particular, this is closely related to results from Ujfalussy & Orbán,  Elife, 2022, which use similar phenomena to support the notion that some form of sampling tied to theta phase oscillations is occurring in the hippocampus.

**Questions:**

Could you elaborate on simplifying assumptions made about hippocampal circuitry? It seems to me that the hippocampus plays no role in inference here (it just relays sensory information as an identity function without any plasticity and plays no role in path integration), whereas in your model the 'unlearned' sensory layer that prevents representational collapse could have been an earlier stage of sensory processing like the thalamus, allowing basal synapses in the hippocampus to learn as has been observed in experimental data (e.g. BTSP).

What happens in this model during 'sleep'? I.e. if there are no vestibular cues provided to this network, does it still sample along its path integrator in a way that preserves spatial information? How does this relate to existing experimental information on sleep 'rehearsal' in the hippocampus?

What happens if more noise is introduced into dynamics of the network during both inference and learning? In particular, I'm concerned about the absence of an explicit gate on plasticity. While it is conceptually an improvement to remove gating of plasticity tied to the theta oscillation, in the model of Bredenberg et. al 2021, the update is not strictly zero for inference parameters during the inference phase--it is only zero in mean (the same is true for the generative phase). As noise grows in the neural output, it is possible for increased variability in these updates, which could longitudinally compound errors or cause some form of representational drift. Have you tested for this possibility? If not, it may be important to note that the gate is only unnecessary in near-deterministic noise regimes.

**Limitations:**

There are no obvious negative societal impacts of the authors' work, and beyond the questions I've raised above, the authors do an adequate job of addressing the limitations of their work and explicitly relating their contribution to previous studies.

---

> ### Author Rebuttal · Authors · 2023-08-09
>
> Thank you for your detailed review, we are glad that you find our contribution “very important” and “clearly written”. We respond to your key points below and including an additional simulation to answer one of your questions. Please kindly inform us if there's anything else required to boost your score.
>
> > ...The authors do not test their learning algorithm in more complex environments. e.g. two-dimensional...[or]... image-based environments.
>
> Please see general response section __2D environments__ for discussion and preliminary results.
>
> Regarding image-based environments we share your suspicions that wake-sleep with no add-ons may not scale well. We made sure not to overstate our claims in the paper. In fact, the reason we chose to model hippocampus, not earlier sensory cortices, is precisely so we can ignore the complexities of these regions and focus on the core computations relating to structure learning in HPC. This is a similar approach to Sorcher et al. (2023), and Whittington et al. (2020). We acknowledge our proposal is only a partial solution and won't account for all learning within the brain.
>
> > The authors don't discuss testable predictions...In particular...that after learning has converged the 'generative' and 'inference' phases of learning 'become indistinguishable,' [which is] unlikely to be true for nondeterministic systems.
>
> Thanks you for suggesting this. We agree and will make this point clearer in the manuscript by removing the word 'indistinguishable'. Of course environmental or neural stochasticity will always cause the generative predictions to “drift” and prevent total convergence. An additional point we’ll add is that we speculate the presence of this drift within each sleep-phase could explain why fast neural rhythms (5 Hz) are used, as they prevent drift accumulating and damaging perception.
>
> Regarding other testable predictions from our model, we list a few here:
>
> * Decoupling HPC from sensory inputs shouldn't immediately damage path integration abilities since this comes from MEC.
> * Decoupling HPC from MEC should damage path integration but not inference.
> * The effect of inhibiting layers in the hippocampal formation optogenetically should depend strongly on which phase of theta we inhibit at.
> * Synaptic learning rules in hippocampal and entorhinal populations may be fundamentally similar.
>
> >Could you elaborate on simplifying assumptions made about hippocampal circuitry? It seems that the hippocampus plays no role in inference...
>
> That's mostly correct although, as we demonstrate in Fig S2, the mapping from sensory to hippocampus doesn't have to be the identity . Also, hippocampus relays information _to_ MEC but also recieves information _from_ MEC and these synapses are all fully plastic. As you correctly identified HPC is not involved in the path integration mechanics which is precisely why the system can "transfer" this ability between environments once hippocampus has learnt to translate the new sensory inputs into the old MEC code. So its role is not trivial.
>
> You're also correct that fixed sensory inputs prevent representational collapse in HPC but they wouldn't prevent collapse in MEC. Since MEC didn't collapse this suggests we could extend the model deeper to include a thalamus which drives HPC via learnable synapses. This is almost definitely closer to the what happens in the brain and would be an interesting future direction. For now we avoided this complexity to focus on our specific goals of answering questions about path integration and structure transfer.
>
> > What happens in this model during 'sleep' if there are no vestibular cues...
>
> In our model, if there were no vestibular (aka velocity) inputs this would be equivalent to path integrating a zero velocity so the bump attractor would stay still. Synaptic noise could, in theory, cause this bump to randomly drift and this drift would preserve spatial information, i.e. it would not jump randomly from place to place. In practice, however, the quasi-continuous nature of our attractor makes each location a very slight local minimum, stabilising the bump against drift. This "stickiness" during path integration was also observed by Vafidis et al. (2021, eLife) and may be a problem for agents attempting to perform path integration at very low velocities.
>
> In terms of how this relates to hippocampal "rehearsal", relevant hypotheses include:
>
> * Large quanities of noise overcome the quasi-continuous stickiness and drive drift during sleep/replay in the absence of vestibular inputs.
> * Top-down processes drive "artificial" vestibular inputs which push the bump around the manifold. This hypothesis was put forward, but not tested, in an opinion piece by Sanders et al. (2015, Trends in Neuro).
> * Feedback inhibition "destabilises" the bump causing it to move of its own accord (Chu et al. NeurIPS 2022)
>
> Our model is well place to answer these questions but, with the time constraints of this rebuttal, we look forward to tackling it in future work.
>
> > What happens if more noise is introduced into dynamics of the network during both inference and learning? In particular, I'm concerned about the absence of an explicit gate on plasticity...
>
> This is an astute observation. One of the core philosophies we take is to avoid hand-crafting learning rules to include terms which may be non-trivial for biology to implement. Explicitly gating plasticity at sub-theta timescales is one such example. Instead we answer empirically, via simulation, whether this term in the theoretical learning rule strictly necessary?
>
> We find that plasticity gating is _not_ necessary. To test this, and in response to your review, we ran the same path integration experiment but with __10 x more noise__. As shown in the attached pdf this results in noisier but still accurate path integration. Only once we increase noise to 100x does learning fails. Thank you for encouraging us to check this, we'll add this result into the supplement.

---

> > ### Comment · Reviewer_Lhyx · 2023-08-10
> > **Response to rebuttal**
> >
> > Thank you for your detailed response. I believe that your response essentially confirms my initial score (7, conf. 4). Good work!

---

> > > ### Author Response · Authors · 2023-08-15
> > >
> > > Thank you again for the fair and thorough review

---

### Official Review · Reviewer_iw5X · 2023-06-23

**Soundness:** 3 good
**Presentation:** 4 excellent
**Contribution:** 1 poor
**Rating:** 4
**Confidence:** 1

**Summary:**

In this work, the authors give a continuous version of the 'impression learning' [1] and use the theta oscillation to modulate wake-sleep phase.

[1] Bredenberg, Colin, et al. "Impression learning: Online representation learning with synaptic plasticity." Advances in Neural Information Processing Systems 34 (2021): 11717-11729.

**Strengths:**

The authors links the theta oscillations to $\theta_t$ in 'impression learning', which gives a neural implementation of the wake-sleep algorithm in Helmholtz machine.

**Weaknesses:**

From a computational perspective, the contribution lacks novelty.

In Seciton 5.2, the authors clearify their relationship to 'impression learning'. They list four differences:

1. continuous version.

The discrete version is equivalent as long as the time steps are sufficiently small. And the authors did not provide the mathematical form of the Evidence Lower Bound (ELBO) for the continuous-time version.

2. $\theta_t$ can be ignored.
3. disregard $1-k_t$.

These are only technical improvments.

4. Link the theta oscillation (5Hz) to $\theta_t$

This is a special case of  'impression learning'.  And the authors did not provide an explanation of how it is implemented from a computational standpoint. They merely established the connection without detailing the computational aspects.

Overall, compared to 'impression learning,' I do not believe this work has enough novelty from a computational perspective. Although it provides a biological explanation of 'impression learning,' I do not think this paper is suitable for the current venue. Perhaps journals like Nature Neuroscience/Communications, or eLife would be more appropriate.





**Questions:**

The latent variable $z$ is not used in the current model, so just disregard it. Eq.(1) can be ignored. If you want to use $z$, there must be a decoder from the neural activity $r$ to $z$.

---

> ### Author Rebuttal · Authors · 2023-08-08
>
> Thank you this review. We’d like to clarify our contributions in case they were misinterpreted. As the manuscript states; “The primary contribution of this paper is to introduce a biologically plausible model of sequence learning in the hippocampus which unifies its capacities as a generative model of sensory stimuli and path integration under one schema”. The intended goal was to shed light on brain function using insights from the ML and not – at least not primarily – to introduce an entirely new model of representation learning (although we make some minor contributions in this regard).
>
> A criticism leveled within this review is that our model doesn't have "enough novelty from a computational perspective". We’d argue this critique slightly misses the point and is oversimplified. Nobody has proposed to model hippocampus as a hierarchical Helmholtz machine nor implemented such a model with learning rules, dynamics and architecture so closely matched to biology, nor shown such a model works. This is a novel computational model.
>
> Perhaps the question comes down to scope rather than significance. We are grateful for your comment recognising this work as suitable for high ranking “journals like Nature Neuroscience/Comms”, however, we also believe that NeurIPS is _the_ preeminent venue for research at the interface of neuroscience and ML and that biologically plausible implementations of key ML algorithms lie well within its scope. A myriad of NeurIPS papers support this claim:
>
> * Pogodin et al., NeurIPS 2023, Towards Biologically Plausible Convolutional Networks
> * Greedy et al., NeurIPS 2022, Single-phase deep learning in cortico-cortical networks
> * Liu et al., NeurIPS 2022, Biologically-plausible backpropagation through arbitrary timespans via local neuromodulators
>
> ...being the most recent. See also:
>
> * Koren et al., 2022, Pemberton et al., 2022, Lipshutz et al., 2020, Bahroun et al., 2021, Clopath et al. 2007, Hasselmo et al., 1994,
>
> ... all published in NeurIPS.
>
> NeurIPS mission statement is to “bring together researchers in machine learning, neuroscience, statistics, …” with listed relevant topics including “Neuroscience and cognitive science (e.g., neural coding)”. Given the comments and ratings from the other three reviewers we are confident that this sentiment is still held within the community today.
>
> In light of this discussion and the additional comments/changes made below we’d be grateful if you’d consider revising your rating and will be happy to attend to changes you consider necessary. Thank you again for the review.
>
> ### Point by point response
>
> > The discrete [and] continuous versions are equivalent...
>
> You're correct, shifting to continuous equations was not fundamental however it allows us to directly compare to real world temporal phenomena e.g. we can say how long in seconds, rather than how many "steps", the path integrator is accurate for.
>
> > authors did not provide the mathematical form of ELBO in continuos-time
>
> Although we agree this is a great suggestion it would be less trivial than it may seem. Unlike, say, Bredenberg et al., we did not derive our learning rules starting from a loss function hence, given the modifications we made, its not guaranteed one would even exist. We'llcarry on giving the discrete formula in the supplement.
>
> > ...are only technical improvements.
>
> These technical improvements are not, themselves, the core contributions of the paper though they may still be considered minor relaxations of past results.
>
> These four point you raised were merely the changes we made to the impression learning architecture for experiment 1. Of course in later figures/experiments we made additional changes which are novel too. These include:
>
> * The addition of self-motion inputs, which is crucial for path integration, to the latent hidden layer.
> * A significant increase in the number of neurons per layer.
> * Training the network on stimuli deriving from a complex and biologically relevant spatial exploration task.
> * Continuous synaptic noise (see methods)
> * ...
>
> > no explanation of how it is implemented from a computational standpoint
>
> We are confident all equations (including learning rates, initialisations, time constants and noise additions) have been detailed in the paper and supplement however if you inform us which equations are missing we’ll be happy to add them. Additionally, section “5.2 Relationship to online Bayesian Inference” connects to our model to impression learning.
>
> > ...I do not believe this work has enough novelty...
>
> To reiterate we consider the core novel contributions of our paper to be
>
> * Creating an interpretable neural model for hippocampal function inspired by previous literature on Helmholtz machines.
> * Successful training of this model in biological regimes to demonstrate it replicates numerous aspect of hippocampal function.
>
> > ...this paper is more suitable for...journals like Nature Neuroscience/Communications...
>
> Thank you, and we take this statement as implying the content of our work is good but that you feel the topic is out of scope for NeurIPS, in which case we respectfully disagree. Please see our above arguments.
>
> > The latent variable z is not used in the current model, so just disregard it...
>
> We had difficulty understanding this comment - if you could clarify we would be more than happy to address this. z is an important variable as it represents the underlying environmental latent so it cannot be discarded.
>
> Thank you for this response. You’re right that not _all_ components of this model’s architecture are entirely novel. This need not mean it lacks computational novelty outright, nor was creating a new ML model ever the intended goal. Even for readers with no interest in biology, bioplausible implementations of ML algorithms are within scope as these lead the way toward novel implementations that better our understanding of how artificial intelligence relates to its biological counterparts.

---

> > ### Comment · Reviewer_iw5X · 2023-08-11
> >
> > **I would like to reiterate and highlight my concerns. Please do not misinterpret my review.** I am not suggesting that neuroscience-related papers are inappropriate for NeurIPS. I also acknowledge that your paper has made some contributions in terms of experimentations. However, compare to 'Impression learning', the novelty of the current paper may not be sufficient for publication at NeurIPS. Assessing novelty indeed lacks absolute standards, but I also wish to quantitatively express my evaluation of the novelty in this paper.
> >
> > The authors summarize the novelty of their paper into two points,
> > 1. Creating an interpretable neural model for hippocampal function inspired by previous literature on Helmholtz machines.
> > 2. Successful training of this model in biological regimes to demonstrate it replicates numerous aspect of hippocampal function.
> >
> > However, 90% of the novelty in the first point comes from 'impression learning'. In fact, the first point should be accurately summarized as 'Making modifications to the impression learning model and using it to interpret the hippocampus.' All these modifications are purely technical, rather than computational. I believe that the remaining 10% of technical novelty is not substantial enough to support a significant argument.
> >
> > Hence, the second point constitutes the main contribution of this paper. However, I perceive the second point as merely an engineering implementation. In other words, this paper falls under the category of an engineering-oriented work, lacking any computational insights for me. I believe that neuroscience articles published in NeurIPS should not lack computational novelty.
> >
> > In simple terms, the computational novelty in this paper has largely been consumed by 'impression learning'. The remaining novelty might be better suited for publication in a journal, rather than emphasizing computational novelty in a NeurIPS context.

---

> > > ### Author Response · Authors · 2023-08-15
> > >
> > > Thank you for your response, we really appreciate you clarifying your comments.
> > >
> > > We strongly disagree. In our paper we generate a new model of the hippocampus by developing and expanding a link with an existing theoretical framework (generative models e.g. Helmholtz machines) - opening the way for a richer understanding of neural computations. We do this using only local learning rules and show the resulting model can learn biological functions such as path integration. We’re pretty sure this is novel and well within NeurIPS’ remit.
> > >
> > > Specifically, we would like to push back on two points:
> > >
> > > 1. __Our modifications to the impression learning algorithm are significant.__ For example the addition of vestibular inputs to the hidden layer of a Helmholtz machine; this is entirely novel and absolutely non-trivial. It was _not_ done in the impression learning paper _nor_, to the best of our knowledge, elsewhere. These are what allow it the learn a tuned path integrator. Likewise the contributions listed above are non-trivial as they empirically prove that paired down bioligical-like learning rules have substantial learning power and can be theortically interpreted in the context of prior work (i.e. generative models). We are happy to do more to clarify the relationship with prior work, but we don’t understand the “10%” statement.
> > > 2. __Even without them, our model is conceptually insightful__. The fact that this algorithm, under the additional elements we have added, yields behavior seen in hippocampus (path integration, bump attractors and transfer learning etc.) is an important conceptual insight, not at all an obvious outcome, and one which we think you may have unfairly disregarded in your review. As we said before, this core contribution is a valuable, novel and well within scope for NeurIPS.
> > >
> > > While our work __does__ make what we consider key methodological contributions, it is worth pointing out that even work that does not introduce novel methods but rather identifies the relationship between proposed methods and neural correlates provide valuable contributions to the field and have previously been considered within scope for NeurIPS. Here's two more examples of highly influential (> 100 citations) NeurIPS publications which applied unmodified ideas from ML to neural systems...
> > >
> > > * __CNNs _with no novel computational modifications_ explains retinal responses to natural scenes__, McIntosh and Maheswaranathan et al. NeurIPS 2016
> > > * __The successor representation with _with no novel computational modifications_ explains the behaviour of place cells and grid cells__, Stachenfeld et al. NeurIPS 2014
> > >
> > > Again we reiterate that we __do believe we make substantial modifications__ but, even if you disagree, hope these serve as counter-examples for why such a hard line on NeurIPS submissions might impede progress. It may even be disadvantageous in the sense that it discourages reuse and consolidation of ideas.

---

> > > > ### Comment · Reviewer_iw5X · 2023-08-16
> > > >
> > > > I will hold onto my perspective; indeed, our interpretations of NeurIPS' novelty criteria may differ. However, since NeurIPS invited me to be a reviewer, I believe it's my responsibility to express my understanding of novelty.
> > > >
> > > > Many machine learning algorithms are indeed inspired by concepts from neuroscience, making it acceptable to use ML models to explain neural system paradigms. However, I am cautious about this approach, as solely relying on existing models can slow down progress in the neuroscience field and limit the emergence of original work. In my view, NeurIPS should feature a greater emphasis on computational novelty.
> > > >
> > > > The papers you mentioned,
> > > > "The successor representation with no novel computational modifications explains the behavior of place cells and grid cells" by Stachenfeld et al., NeurIPS 2014, seems to be a discrepancy. I could only find "Design Principles of the Hippocampal Cognitive Map" by Stachenfeld et al., NeurIPS 2014. Although their model also used well-developed ML techniques and lacked a consideration for biological plausibility, its novelty stemmed from introducing RL **concepts** to the Cognitive Map paradigm. As the concept of generative models is widely accepted in hippocampal modeling, I expected your work to have model-level (**implementation**) novelty. Unfortunately, the use of the impression learning model in your work didn't showcase such novelty.
> > > >
> > > > Furthermore, I have two questions about your work:
> > > >
> > > > 1. How does the LFP theta frequency oscillation modulate your model's $\theta$, as represented in Eq. (4) & Eq. (5)? Specifically, how is this modulation implemented in the neural circuit? Your paper lacks substantial discussion in this area, which is a core assumption of your work.
> > > >
> > > > 2.  Animals exhibit theta oscillations mainly during wakeful and active states, while sharp waves and ripples dominate during other times, often associated with replay events. How do you explain these experimental phenomena in the context of your model?
> > > >
> > > > If you can answer these two questions, I promise to raise your score.

---

> > > > > ### Author Response · Authors · 2023-08-16
> > > > >
> > > > > > I will hold onto my perspective; indeed, our interpretations of NeurIPS' novelty criteria may differ. However, since NeurIPS invited me to be a reviewer, I believe it's my responsibility to express my understanding of novelty.
> > > > >
> > > > > Of course! And we appreciate you expressing your opinion; comparing novelty between works is difficult, subjective and, like you said, lacks absolute standards.
> > > > >
> > > > > > How does the LFP theta frequency oscillation modulate your model's $\theta$ as represented in Eq. (4) & Eq. (5)?
> > > > >
> > > > > $\theta$ in our model _is_ the LFP theta oscillation. We approximate this using a square wave switching between 0 and 1 (Eq. (5)) with a time period of $T$, the inverse of the LFP theta frequency, $T = 1/f_{\theta} = 200$ ms (as stated on the line below it 146). Does this answer your question?
> > > > >
> > > > > > Specifically, how is this modulation implemented in the neural circuit?
> > > > >
> > > > > The modulation is directly implemented by Eq. (4) as a "gating" controlling which dendritic compartment drives the soma depending on the (time-varying) value of $\theta(t)$. If $\theta = 1$ -- the first half of the cycle -- then only basal voltages will gate into the soma ($p = p_B$). If $\theta = 0$ -- the second half of the cycle -- only apical voltages will gate into the soma ($p = p_A$). This local gating has network level consequences: when $\theta = 1$ information flows up the hierarchy ($z \rightarrow p_B \rightarrow p \rightarrow g_B \rightarrow g$) whereas when $\theta = 0$ information flows down the hierarchy ($g_A \leftrightarrow g \rightarrow p_A \rightarrow p$). This amounts to sleep-wake cycles.
> > > > >
> > > > > We would hypothesize this gating is implemented by bio/electrochemical processes at the intra-cell level which we do not model here. Though we don't model to this level of complexity, it has been measured experimentally that timing between opposing dendritic inputs alternates in this manner at the theta frequency (see Fig. 2, Bittner et al. 2015, NatNeuro). We'll bring this observation forward in the paper to clarify.
> > > > >
> > > > > > Animals exhibit theta oscillations mainly during wakeful and active states, while sharp waves and ripples dominate during other times, often associated with replay events. How do you explain these experimental phenomena in the context of your model?
> > > > >
> > > > > This is wonderful question and one we also touched on in our rebuttal to reviewer Lhyx which you may additionally like to check out. In summary, we believe the bulk of learning occurs during wakeful active states when theta, and therefore our mechanism, is active. This is also when the animal is actively doing path integration. Ripples, replay and sharp waves are likely _additional_ phenomena implicated in different processes e.g. memory consolidation and/or planning. This does not mean they couldn't interplay with our model: these event could derive from spontaneous or driven activity on the MEC bump attractor. If, for example, during a sharp wave ripple internal noise or artificial vestibular inputs drove the bump attractor rapidly this would trigger a replay event downstream in hippocampus and potentially further learning (this is just one propsal, there are others).
> > > > >
> > > > > Lastly spiking during ripples (replay) is subject to a form of phase precession but relative to the ripple band oscillation (150-200Hz) as oppose to the theta (Bush et al., 2022, Current Biology) - this actually suggests that all of the mechanisms we propose would work albeit at a totally different timescale (meaning learning rules etc would have to be adjusted to match), though we don’t yet have enough biological data to know if this is a plausible scenario.
> > > > >
> > > > > We hope these suffice to answer your questions and thank you for offering to increase your score - if anything is still unclear please don't hesitate to ask.

---

### Official Review · Reviewer_UjUg · 2023-07-02

**Soundness:** 3 good
**Presentation:** 4 excellent
**Contribution:** 2 fair
**Rating:** 7
**Confidence:** 3

**Summary:**

This work presents a neat model that incorporates aspects of hippocampal function under one umbrella:
* first, the input from the environment (z) goes into the sensory layer (p) and activates the internal state (g) in a certain way, the model captures this an "inference" or "wake" stage of the training
* next, a set of recurrent connections in the "internal state layer" g simulates the prediction-making, guessing the next internal state, simulating the predictive mechanism in navigation (or potentially other cognitive functions)
* finally, a generative pathway goes the opposite way from g down to p and attempts to generate what the sensory input should look like

Training of the inference and the generative parts of the model is alternative with a 5 Hz square wave function, which reflects existing observations of the role of this oscillation in hippocampus.

The contribution of the paper is just to propose a model that describes the dynamics of simple dynamical processes, with potential extension to path integration.

------------- Update after the rebuttal period -------------

I would like to thank the authors for a very detailed rebuttal and being engaged in the discussion on both the technical and the ~philosophical levels.

My main criticism was based on the idea that just training a model that captures something is not enough, because there is an infinite number of models & architectures that will be able to do that. However, after the rebuttal period and skimming the paper again I came to think that the model presented in this paper achieves more that just capturing the dynamics by "any means necessary", but actually does so under heavy restrictions, and, while there are still multiple such models possible, the fact that the model still works under said restrictions makes this result interesting.

I am raising my score to "7: Accept" as this work, in my estimation achieves exactly the required level of "high impact on at least one sub-area" (computational modelling of hippocampal formation) and has "good-to-excellent evaluation, resources, reproducibility".

**Strengths:**

This is a well-written paper, it has a good flow and is quite understandable. The proposed model combines various ideas about brain function in an elegant way and, while remaining simple, does manage to make those ideas to work in unison. The relevant work section gives great context for the work. The experimental work is well explained and documented.

**Weaknesses:**

As someone who is not coming from the attractor perspective I fail to understand the significance of an attractor appearing. In my mind (please help me understand why this is not the case) the appearance on an attractor is exactly the goal and purpose of training. Basically, the way I see it, if we have a certain dynamical process, and we have successfully trained a neural network to capture this dynamics (the model can predict s(t+1) from s(t)) then saying "it formed an attractor" is equivalent to saying "the learning has converged". But isn't that precisely the purpose of training the model? The mere fact of it converging (and forming an attractor that captures the training data trajectory on a manifold) is, of course, a good thing -- the learning was successful, but is not in any way unexpected or remarkable. After all this is precisely what we wanted to happen -- we make an effort to build a model and the learning process that captures the data, and if done correctly, that's exactly what it is going to do.

In this work the fact that the generative part of the model had formed an attractor (aka "can generate the dynamical process correctly") is presented as a significant outcome. But from the machine learning perspective this is a trivial result, that IS what happens when you train something successfully and is not sufficient to imply a special biological significance of the model.

Put it this way: if I would make this model to have not 1 sensory and 1 hidden layer, but, let's say 3 sensory and 4 hidden layers -- it would, of course, also form some sort of an attractor (with a more complex underlying manifold and shape due to the fact that we have more layers), but that would not tell me anything about this model's biological plausibility.

I guess what I would like to discuss with the authors and other reviewers is whether creating A machine learning model has scientific value, or (as I posit) we need to make a step further and offer a model that would satisfy more than just capturing the dynamics, but also make predictions or coincide with biological restrictions that we did not explicitly encode, etc. Then -- yes -- we could say that not only this model captures the dynamics, but it also the only variant (or at least belongs to a small family of variants) that also do X, Y and Z the way the brain does it.

In the olden days of modeling proposing a set of differential equations that capture the dynamics was impressive because it was not a given than such a set of equations can ever be found. The outcome of such scientific endeavor could fail if the scientist was not able to describe the dynamics. So when a model was found nonetheless it made the contribution significant. But can we really apply the same criteria to ML-driven modeling? Because in the case of ML it is almost a given, that a "set of equations" (now represented by an artificial neural network) will be found to satisfy the data. Which tells us nothing more that "this data can be described" (which is rather noncathartical) and the is likely to be an infinite amount of such models that would do so (even the same architecture, trained again from a different initialization starting point would likely produce a "new" model).

I am looking forward to discussing this with other reviewers and the authors, to help me understand (1) am I wrong about the (in)significance of an attractor appearing and (2) whether it is too much and prohibitive for science to ask more from a model in order to count it as a significant contribution to the field than just it being able to capture the data, especially when this data is simulated?

For the authors, one way to make this critique constructive and actionable I guess would be to say more in the paper about why this particular model (it's architecture and other characteristic) is especially suited to described hippocampal dynamics and why all the same conditions would not be satisfied by any number of similar models.

Other points:

* It would be great to see some ablation experiments that would help understand that the observed experimental results are unique to the proposed model and would not emerge if some of the critical components of the model are turned off (and help assess their criticality).

* I also ask this as a question below, but I wonder why 2D environments were not explored experimentally? It would add so much to the work, make it relevant to more people and help build connections to existing work on hippocampal function, in particular place/grid cells.

**Questions:**

Fig 1: What would be the appropriate intuition for the g layer? If z is "input" and p is "sensory", then should we think of g as an "internal state"?

95: Are there studies that support the idea of the theta rhythm acting as the direction switch? Are there recordings that somehow show the change in the direction of the flow of information?

147: Although in the Introduction (line 102) you mention that there is not discrete time assumption and the whole thing works continuously, I still can't help by wonder how many weight updates "fit" into one 5 Hz cycle for "wake" and "sleep" steps? Is this number driven by the computational power of the simulation hardware, cramming as many updates as possible, or there is a biologically-informed constrain that says something like "within one theta cycle synaptic connection can be updated only X times"? How does your implementation of the model handle this? Note: while the formulation is continuous, the act of updating is still a discrete step that takes place at some point, and this is what my question is about, basically how many times within one cycle does the line of the code that does w = w + update is being executed?

156: Is it correct to think of g->g recurrent connection as a step in the internal representation space where the brain tries to predict what the next (t+1) state will look like? And this internal representation is then used as a starting point for the generative part of the model to correctly generate the state of the environment z_t+1?

161: What is the "ground truth" for training w_p_A and w_g_A in the wake cycle? How do we know what what we should train toward do? I understand that the training is local and works to minimize the differences between dendritic (apical for the wake cycle, right?) and somatic activation, but how do we know that the dendritic activations represent the "true" ones and that bring the the system closer to them will result in successful learning?

161: Same question rephrased: what is considered to be correct output of the inference part of the model?

164: What is the ground truth of the generative training step? Let's say I run 1 pass from g -> p -> z, where does the learning signal come from? Do we compare the generated z_t+1 with the actual one to know whether the generative step has generated the correct stuff? What is considered the "correct" output of the generative part of the model and what this output is compared with in order to be able to say "yep, it has generated correctly"?

209: When you say that only the "wake" part of the model is activated, how do you conclude that is "correctly autoencodes"? I guess this question is still stemming from the ones above, but I fail to see what is considered the ground truth? Does it go all the way to z, or it is sufficient at this point to train w_g_B (as per Fig 1) to produce such a g_t, that after g->g step that will create g_t+1 this g_t+1 will correctly be reconstructed via w_p_A back into original p?

209: Another question here is when you "autoencode" do you autoencode p_t into p_t or do you "autoencode over time" so that p_t autoencodes into p_t+1? If not the latter, then I am confused when does the g->g step happen and what's the intuitive meaning of that step.

(Sorry for the avalanche of questions, I imagine these will be hard to parse, I tried to formulate them the best I could :) )

233: If the hippocampal "internal" part of the model is trained to autoencode g->p, then why would we expect sensory lesion to affect that in any way? You say it is remarkable that is keep generating, but isn't it exactly the behavior we would expect? I am trying to understand why the fact that g->p keep working is "remarkable" when we turn the z off. Since z is not part of that machinery, isn't it trivial that g->p will keep doing it's thing? Please let me know what I am missing and if the observation is not trivial as I postulate, then would would be the trivial behavior that we would expect but that is remarkably not occurring?

280: It would be interesting to confirm that indeed MEC has learned some basic property of path integrating by running an ablation experiment, where MEC is randomly initialized and for 30 minutes on the MEC <-> HPC is allowed to train. As with echo networks it might the case that learning MEC <-> HPC is always sufficient even is MEC internal state is random. In the ablation experiment you can check whether using pre-trained MEC is any different from using randomly initialized MEC (given the same approximate shape of the value distribution), would it result in longer re-training times, worse performance, etc or not.

301: Would anything change (in terms of the attractor, performance, etc) in instead of 5-10 Hz oscillations between wake and sleep you would use 1 Hz? 20 Hz? Is the 5 Hz crucial to the success of the model, and if yes - what would "break" is another rhythm would be implemented?

* What was the main reason for choosing very simple experimental environment for this work? A 2D navigation experiment would help show so much more and build empirical connection to place/grid cells.

**Limitations:**

There is not discussion of the limitations in the paper, and while I personally do not consider it a must-have, in this case it would be useful to help understand how far-reaching the authors deem their clams and results and which aspects of it rely on strong assumptions.

---

> ### Author Rebuttal · Authors · 2023-08-08
>
> Thank you for this incredibly thorough review. Your insightful comments have led to meaningful improvements. We respond point-by-point below but these first paragraphs are reserved to further your philosophical discussion about the goals of computational modeling.
>
> ### General response about the merits of computational modeling.
>
> In ML, models are “trained” to minimize an objective function almost always with some type of gradient descent. In this case, users “put in” to their objective functions exactly what they want, and often manage, to get out. As you said, in the olden days this was merit-worthy but less so today with the powerful algorithms/architectures available. Even language comprehension is arguably a solved problem.
>
> On the other hand, in computational neuroscience people often describe “learning” which, at a coarse level, involves more simple learning rules reminiscent of those ubiquitously observed across the brain. End-to-end optimization of objectives with backpropagation is considered implausible.
>
> We are strongly in the "learning" regime. Obviously with the right objective function and optimizer we could have achieved the same results. A network trained this way would perform better but it would be sufficiently trivial that, without further justification, we wouldn’t have submitted it to NeurIPS. In our case we view the attractor and other results of learning as _emergent_ properties. As we’ll elaborate in the point-by-point it was not the only possible solution and, although as scientists we were guided by intuition to suspect this solution, it was in no way “built in” to the optimisation procedure.
>
> Of course an objective function (sort of) exists – in our case we liken the system to a Helmholtz machine optimizing an ELBO-style loss – but that does not make the solution trivial since the learning rules, architecture and dynamics with which we (approximately) optimize this are heavily constrained by biology: No back prop, no long memory traces, no non-local information. The primary value of our paper is therefore demonstrating that the learning rules and architecture of HPC can be seen through the right mathematical lens as roughly the same as a well studied machine learning system (we don't think anyones done this before). To put it another way, we took a step towards discovering the objective function of HPC and (to us at least) this finding is profound; speaking to deeper organizational principles at play within the brain.
>
> All said, we believe we are in firm agreement with you: the value is absolutely not in a model converging but in what this teach us about the brain. We believe the goal of modeling should be to (i) unify previously disparate concepts and (ii) make testable predictions. On both accounts we believe we are achieving this goal. Thank you again for opening the discussion on this important topic. If you agree we’d be happy to carry on the discussion below and if it’s made you reconsider the contribution of our paper we’d be grateful if you might revise your rating to reflect this.
>
> We'll now respond point-by-point but the character limit means we won't fit all your questions in this box. We hope it is ok if we finish responding to your questions in a comment which we'll post after the rebuttal period closes.
>
> ### Point by point response (part 1/4)
>
> > ...I fail to understand the significance of an attractor appearing....an attractor is exactly the goal and purpose of training....
>
> Perhaps it would help to begin by clearly stating a question which, prior to our work, we do not believe there would have been consensus within the field about the answer to: _“Can unsupervised local learning in a system with one hidden layer receiving unstructured spatial inputs learn transferable path integration or is back propagation through time and/or deeper non-linear architectures a requirement?”_
>
> State-of-the-art before us was that relevant HPC modelling almost always used non-local learning rules and powerful optimization algos which find path integration / attractors as, in your words, this was “precisely the purpose of training” (e.g. Sorscher et al. 2023 and Banino et al. 2018). It's unclear if this was because authors weren’t interested in biological plausibility or if its because it was _necessary_ for successful training.
>
> To reiterate our earlier point we don’t just put in what we get out. We dont just do literal gradient descent on a hand-crafted objective function. The appeal of the attractor is exactly that it emerged from a system so heavily constrained by the realities of biology, lending weight to the suggestion that our model may contain some truths about the brain.
>
> An observation from Fig 3c may help: notice a subset of neurons have converged on a trivial but perfectly valid solution of zero weights. They “died”. Their existence neatly demonstrates that a functioning attractor manifold over all/most hidden units is not _the_ solution but rather _a_ solution. We are the first to demonstrate this solution can emerge under the constraints used. Also, (see points further down) the system doesn't predict $s(t+1)$ from $s(t)$ but $s(t)$ from $s(t)$ making the future-predictive nature of the attractor more remarkable.
>
> This is why we respectfully disagree with the statement that
>
> > saying "it formed an attractor" is equivalent to saying "the learning has converged".
>
> Sure, learning converged. But nothing a priori guaranteed an attractor would be the fixed point.  In the remaining space we will add sentences to clarify our interpretation of this result and also reword instances where we have lent too far toward the interpretation that convergence alone is the key take home result, which it is not.

---

> > ### Author Response · Authors · 2023-08-10
> > **Point by point response (part 2/4)**
> >
> > > ...[a model should] make predictions or coincide with biological restrictions...
> >
> > You’re absolutely right that dissimilar models can explain the same variance or data. The difference here is the strong correspondence between the elements we use to build the model and the known biology of hippocampus. This model does make unique testable predictions, here’s a couple:
> >
> > * Decoupling HPC from sensory inputs should not immediately damage path integration abilities since this comes from MEC.
> > * Decoupling HPC from MEC should damage path integration but not inference.
> > * The effect of inhibiting layers in the hippocampal formation optogenetically should depend strongly on which phase of theta we inhibit at.
> > * Membrane potentials at basal and apical dedrites should converge over time.
> > * Synaptic learning rules in hippocampal and entorhinal populations may be fundamentally similar.
> > * ...
> >
> > None, or not all, of these are made by other models published before now.
> >
> > Unrelated to neuroscience, studying the brain’s implementation of ML algorithms  can contribute beyond data-fitting by shedding light on how it can learn in a more data- and energy-efficient manner than most large-scale ML algorithms, potentially leading to improvements.
> >
> > > ...Discuss whether (1) am I wrong about the (in)significance of an attractor appearing and (2) whether it is too prohibitive for science to ask more from a model than it being able to capture data
> >
> > (1) Your point is well taken but we respectfully disagree. We think the appearance of an attractor is significant given the constraints on the system within which it emerged.
> >
> > (2) Mostly we agree but perhaps propose that there is a sliding scale, for example people can now answer a qualified “yes” to the question we posed above and use this to move discussions forward. Our model makes predictions as listed above and in the manuscript. We are already looking towards testing this model on real data as part of ongoing work.
> >
> > > ...one way to make this critique constructive and actionable...
> >
> > We’ll highlight and extend sections where we have done this and add additional sentences directly contrasting our model to others for the camera ready. Thank you for this constructive suggestion.
> >
> > > It would be great to see some ablation experiments...
> >
> > Some ablation studies have already been performed to stress test the model
> > * Lesioning of sensory inputs (Fig 3).
> > * Relaxing the constraint that HPC receive unimodal spatial inputs (Fig. S2b)
> > * Relaxing the identity constraint on weights from MEC to the conjunctive units (Fig. S2c)
> >
> > In the attached pdf you’ll find results for three new ablation studies, summarised in the general response section __Additional simulations__. These were performed in response to your review. Thank you for this suggestion which has led to a genuinely interesting set of new results. We will include these (and any others you feel are important) in the supplement.
> >
> > > ...I wonder why 2D environments were not explored...
> >
> > See general response __2D simulations__ for discussion and preliminary results.
> >
> > > the appropriate intuition for the g layer
> >
> > “Internal state/latent” would be a fair interpretation of $g$. We’ll clarify this in the paper.
> >
> > > Are there studies that support the idea of the theta rhythm acting as the direction switch
> >
> > Yes, some. Although recall from Fig. 1 that information flows bidirectionally _at all times_ in our model. The "direction switch" happens internally within the neurons and effects which direction information flows contiguously through the hierarchy. All synapses are still active at all times and this would make measuring the switch directly hard. Papers/results linking theta phase to the direction of information flow include:
> >
> > * Analysis of theta phase precession plots shows they are bimodal indicating that the first and second half of the cycle are used for different computations (Yamaguchi, J.Neurophysiology, 2002)
> > * Hasslemo (2002 Neural Comput. and 2014 Neuroimage) showed that hippocampal dynamics are consistent which alternate phases of encoding ($\sim$“wake”) and retrieval ($\sim$“sleep”).
> > * Sanders et al. 2015 proposed (though didn’t computationally test) a similar idea that distinct phases of theta are separately used for inference and prediction.
> > * Ujfalussy & Orbán, Elife, 2022 (pointed out by another reviewer).
> >
> > We’ll add the above citations which weren’t in the manuscript already.
> >
> > > ...how many weight updates "fit" into one 5 Hz cycle for "wake" and "sleep" steps?...
> >
> > The answer is $\frac{T_{\theta}}{dt} = \frac{200ms}{25ms} = 8$ but there is nothing fundamental about this number. The shift from discrete to continuous equations was also not fundamental (as you say, we discretise eventually at the level of simulation) but does allows us to directly compare the system and its dynamics to real world temporal phenomena e.g. we can measure how long the path integrator is accurate for in seconds not "steps" which feels useful.

---

> > ### Author Response · Authors · 2023-08-10
> > **Point by point response (part 3/4)**
> >
> > > [are] g->g recurrent connections [trying] to predict what the next (t+1) state will look like?
> >
> > Yes. The $g$->$g$ weights update the internal representation from $g(t)$ to $g(t+dt)$ as given in Eqn. 3. A fuller version (Eqn. 10) reminds readers that Eqn. 3 merely approximates  a continuous dynamical system in which case the $g$->$g$ weights are better understood as the parameters of the function determining the _rate_ at which $g(t)$ is changing.
> >
> > > What is the "ground truth" for training $w_{p_A}$ and $w_{g_A}$ in the wake cycle
> >
> > We don’t know! That’s partly why this is interesting. Each layer just “locally” (or "greedily") optimises its own synapses in a way which is not _necessarily_ optimal for the other layers.
> >
> > As already mentioned another stable solution under these local dynamics is when $w_{g_A} = w_{g_B} = 0$. The entorhinal system could die/collapse which would certainly be undesirable for the hippocampal layer below it. A better solution is when the non-zero recurrent apical predictions in $g$ match the non-zero basal predictions (arriving from $p$) but, as you correctly identified, this system is not sufficiently constrained to determine the one and only final solution for $w_{g_A}$.
> >
> > There is some weak cross-talk between layers. Concurrently, during wake, the sensory layer $p$ wants to match its apical (from $g$) to its basal (from $z$) inputs. If $g$ is a lousy latent representation the system can't achieve this and $w_{p_A}$ will continue learning. This can then indirectly affect $g$ since $w_{g_B}$ will be trained during the subsequent sleep cycle on inputs from $p$ which ultimately came through $w_{p_A}$. It is possible that this weak and indirect interaction between learning in $p$ and learning in $g$ is what allows a global solution to be found since both layers keep learning until stability (if this ever occurs) but nothing guarantees or enforces their cooperation.
> >
> > Compare this to backpropagation where interlayer updates are connected via the chain rule and layers effectively “cooperate” towards a mutually optimal solution. Not so with local learning and the emergence of mutually compatible representations supported by calibrated attractor manifold is a non-trivial result.
> >
> > > What is the ground truth of the generative training step?...What...output [allows me to say] "yep, it has generated correctly"?
> >
> > Again, there is no “ground truth” in the way the term is used in supervised learning. There are only representations ($z$, $p$ and $g$) and mappings between those representations ($w_{p_A}, w_{p_B}, w_{g_A}, w_{g_B}$) which preserve information.
> >
> > Here's a more simple answer (but please don't presume the learning rules are directly optimizing for this solution). In a linear system the “yep, it has generated correctly” moment would occur whenever $w_{g_B}$ and $w_{p_A}$ are pseudoinverses of one another. In our case there is a non-linearity to account for as well but the intuition holds.
> >
> > To answer your question more fully, figure 1 panel c may help: during the generative “sleep” phase the weights which will be updated are $w_{g_B}$ which map the sensory layer $p$ to $g$ (through the dashed-green arrow). During this sleep period $p$ is receiving top-down drive from $g$ (through the thick blue arrow). Thus the generative step is entirely self supervised! During sleep MEC is decoupled from reality. It is dreaming and learns from these dreams. But - as is normal for wake-sleep algorithms - any learning which occurs during these dreams constrains learning during subsequent awake (see comment above) and so on so forth until (hopefully) convergence. We appreciate the oscillatory nature of this model be a bit mind-bending at times!
> >
> > > ...how do you conclude that is "correctly autoencodes"?
> >
> > We claim it “correctly autoencodes” the input if the sensory input $p_B$ is matched by the prediction $p_A$ (when the system is forced into a "wake" cycle. If the system were linear this would be satisfied if $w_{g_B}$ and $w_{p_A}$ were pseudoinverses of one another so the mapping from $p$->$g$ is undone by the mapping back from $g$->$p$. We will add to the manuscript the pseudoinverse intuition for autoencoding to clarify.
> >
> > > ...do you autoencode p_t into p_t or do you "autoencode over time" so that p_t autoencodes into p_t+1...
> >
> > It is the former. i.e. it’s same-step prediction ($p_B(t) = p_A(t)$). Sorry if the phrasing “over time” confused things, we will reword.
> >
> > The $g$->$g$ step happens all the time but it’s only during sleep that it becomes important. The key is to remember the compartmental nature of the neurons.  During the wake phase, sure, $g$->$g_A$ is _active_ but this then stops at the dendritic compartment $g_A$ and does not gate into the soma, hence cannot impact anything further down the hippocampal hierarchy. Only during sleep does $g$->$g_A$ becomes important as this then gates back into $g$ and down into the $p$.

---

> > ### Author Response · Authors · 2023-08-10
> > **Point by point response (part 4/4)**
> >
> > > Sorry for the avalanche of questions, I imagine these will be hard to parse...
> >
> > So far so clear! We’re happy to answer them.
> >
> > > If the hippocampal "internal" part of the model is trained to autoencode g->p, then why would we expect sensory lesion to affect that in any way?
> >
> > You’re absolutely right and have not misunderstood. $z$ is not part of the generating machinery but, before lesion, acts as a calibrating signal returning the attractor to the correct place every theta cycle. Therefore, back to our main point, what is “remarkable” (although we will reword this for the camera-ready) is that an accurate path integrator emerged in the first place.
> >
> > > It would be interesting to confirm that indeed MEC has learned some basic property of path integrating by running an ablation experiment.
> >
> > We did exactly this experiment (and it failed, showing MEC learning is important). See general response section __additional experiments__.
> >
> >
> > > Would anything change if instead of 5-10 Hz oscillations between wake and sleep you would use 1 Hz? 20 Hz?
> >
> > In a related study, Bredenberg et al. (NeurIPS 2021) have tested the impact of increasing or decreasing sleep-wake cycle durations. In summary it doesn’t matter what frequency you train in terms of how fast the learning will converge but the biggest effect comes when you analyze the online dynamics. This lower bounds the desired frequency: If the wake-sleep cycles are too slow then prediction accuracy during sleep can quickly “drift” off as the bump attractor inevitably decalibrates. This is disadvantageous for a system which wants to maintain accurate perceptions at all times during behaviour. There may be an upper bound to the frequency in terms of the temporal dynamics of synaptic plasticity or neuronal membrane timescales (neither of which are modelled in detail here) which require clean distinguishable sleep-wake cycles. One speculative conclusion is that 5-10 Hz is _just fast enough_ to enable accurate perception at all times but _just slow enough_ to satisfy the dynamic constraints.
> >
> > > A 2D navigation experiment would help show so much more and build empirical connection to place/grid cells.
> >
> > We are looking into this for future work, please see general response section __2D environments__ where we have some positive preliminary results.
> >
> > __Thank you__ again for this excellent review which has prompted a lot thought. We hope we have satisfactorily answered all your questions and made the suggested additions to improve the manuscript. If it has made you reconsider our papers contribution we'd be grateful if you'd consider revising your score to reflect this or let us know what else might be required.

---

> > ### Comment · Reviewer_UjUg · 2023-08-20
> >
> > Hi everyone, sorry for the delay, I will now work through the responses and comment as as I move along.
> >
> > > the system doesn't predict s(t+1) from s(t) but s(t) from s(t) making the future-predictive nature of the attractor more remarkable.
> >
> > Indeed, this is an important clarification! It does put significant brakes in my objection that you get out what you put in.
> >
> > Another thing that I have not appreciated enough how local the update rule is. Will comment on this below.

---

> > ### Comment · Reviewer_UjUg · 2023-08-20
> >
> > Ok, the responses do not thread under each box, so I might as well put all together into one.
> >
> > Response part 3/4 made me realise that the locality of learning more crucial to everything and it is what makes many of the following observations non-trivial. To help deliver this to other readers (who, one might hope would read carefully, but you never know) -- perhaps it you could split 2.2 into two separate subsections, one that explains the local rules and how those are different from end-to-end, and the second about gating with theta oscillations?
> >
> > Thank you for such a detailed rebuttal! Your responses and additional experiments have improved my understanding a lot, I will re-read and re-evaluate with this new appreciation for non-triviality of some of the outcomes.

---

> > > ### Author Response · Authors · 2023-08-20
> > >
> > > Thank you for responding, we're glad our rebuttal was clarifying. It's our job to make this clear to the reader and we agree that dividing section 2.2 will help clarify the results and explain their "non-triviality" in comparison to non-local learning so we're happy to make that change.
> > >
> > > Once you've had a chance to rerate our work please let us know if there are any further questions so we can respond before the end of the rebuttal period!

---

> ### Author Response · Authors · 2023-08-15
> **Follow-up**
>
> Hi, just wanted to make sure you didn’t miss this. Let us know if you have any additional questions, we'll be happy to answer.

---

### Official Review · Reviewer_PiE7 · 2023-07-07

**Soundness:** 3 good
**Presentation:** 3 good
**Contribution:** 3 good
**Rating:** 6
**Confidence:** 4

**Summary:**

The Hippocampus is postulated as a generative model to learn latent state representations and generate sensory predictions to solve spatial and nonspatial tasks. Theta-band oscillations are used to gate information flow into the generative model to modulate learning. A ring attractor develops within the generative model for path integration and flexibly transfer structures between environments. These computations are performed using biologically plausible learning rule.

**Strengths:**

-	Scheduling neural dynamics with theta oscillator: Model neurons are usually described as input summation followed by a nonlinear transformation. Here, the authors expanded the computational dynamics of a neuron by coupling the membrane potential with the theta oscillator (Eq. 4). This affords fine-grained control or better scheduling of the neurons and plasticity using a central gating mechanism.

-	Biologically plausible learning rule: The model seems to be an extension of the MESH architecture which uses the Pseduoinverse learning rule. Here, the network is using Hebbian like learning algorithms to learn a ring attractor for path integration and can dynamically relearn a new environment even with some of the weights are fixed, adding to the list of biologically plausible models.


**Weaknesses:**

-	Not a completely novel architecture: Although the multi-compartment neuron architecture controlled by the theta oscillator is novel, past works have described similar neural architecture (Sharma et al., 2022; Han et al., 2020) of having two hierarchical layers recurrently connected. I am curious to know how these neural architectures correspond to the author’s proposed architecture and if there can be some overlap in learning performance?

-	Insufficient analysis: It is not clear what computations the generative model performs using the ring attractor. The inference model seems to resemble place like receptive fields in Fig. 4b but so does the generative model? I anticipated grid like fields instead but this is not the case? It might also be interesting to show the synaptic weights ($Wgb$, $WpA$) evolve through learning and offer some insights to its computation, similar to Fig. 3c.

-	Inclusion of network prediction prior to learning: Could the authors include the y axis information in Fig. 2b to show how much of the prediction error has been reduced through learning, and the activity of the inference (Fig. 2c) and generative (Fig. 2d) model prior to training to determine the difficulty of the task and the effectiveness of learning?

-	Path integration capability: Fig 3e demonstrates that the mean decoding error increases monotonically beyond 20 cm by 5 seconds. Can this high error still be claimed to be accurate path integration, where prior to the lesion, error was almost 0? What else could be integrated into the model such that the error does not increase as fast as in Fig. 3e?

-	Application to a 2D navigation task: Authors demonstrated the model’s application to a 1D navigation task (left and right) which is rather limiting. Learning to path integrate in a 2D environment will be much more convincing. Furthermore, if the authors could demonstrate the agent ability to perform vector-based navigation by estimating its location using path integration and recalling a goal location from a memory system (Kumar et al., 2023; Foster et al., 2000), this will be a complete contribution to the research in biologically plausible spatial navigation using the entorhinal-hippocampal model.

-	Insufficient ablation studies: Ablation studies of the theta oscillator, multi-compartment neurons and individual layer plasticity will give us a better understanding of the roles played by each of these components for learning a ring attractor for path integration and remapping to a new environment.

References:

Han, D., Doya, K., & Tani, J. (2020). Self-organization of action hierarchy and compositionality by reinforcement learning with recurrent neural networks. Neural Networks, 129, 149–162. https://doi.org/10.1016/j.neunet.2020.06.002

Sharma, S., Chandra, S., & Fiete, I. R. (2022). Content addressable memory without catastrophic forgetting by heteroassociation with a fixed scaffold. ICML. http://arxiv.org/abs/2202.00159

Kumar, M. G., Tan, C., Libedinsky, C., Yen, S., & Tan, A. Y.-Y. (2023). One-shot learning of paired associations by a reservoir computing model with Hebbian plasticity. ArXiv. http://arxiv.org/abs/2106.03580

Foster, D. J., Morris, R. G., & Dayan, P. (2000). A model of hippocampally dependent navigation, using the temporal difference learning rule. Hippocampus, 10(1), 1–16. https://doi.org/10.1002/(SICI)1098-1063(2000)10:1<1::AID-HIPO1>3.0.CO;2-1



**Questions:**

-	Would the plasticity rule still work if it was purely Hebbian plasticity i.e. $dW/dt = p(t)*g(t)$ without the additional normalizing term?

-	The authors contrasted the wake-sleep algorithm to backpropagation. Could the authors compare the intuition of the wake-sleep algorithm to contrastive Hebbian learning? This might be a way to describe contrastive Hebbian learning using a biologically plausible alternative.


**Limitations:**

The authors have sufficiently addressed their future directions. Model limitations and conditions at which the model breaks could have been further explored.

---

> ### Author Rebuttal · Authors · 2023-08-08
>
> Thank you for this detailed review which has led to a number of changes to the paper. We have addressed your comments and questions below but please respond with any additional questions which we’re happy to answer.  Apologies if our answers seem at all "curt", we are heavily constrained by the 6000 character limit but would like to cover all your points.
>
> > Not a novel architecture....Sharma et al., Han et al.,...
>
> We weren't familiar with the work of Sharma et al. but agree their architecture is similar, including its relationship to HPC-MEC. Two differences stand out: firstly we focus on generative modelling of the temporally varying environmental stimuli but they focus on content addressable memory of static stimuli. Additionally we use multicompartmental neurons which "multiplex" top-down and bottom-up signals and a biologically plausible learning rule. It is exciting to see these architectures applied to hierarchical RL as in Han et al. Comparing learning performance between these models would be quite non-trivial as all three networks perform substantially different tasks (generative modelling, memory and RL). We'll add these references.
>
> Whilst there are numerous other examples of two layers recurrent architectures in ML we'd debate whether this is a "weakness". The combination of its features and results make our model novel even if the architecture alone is inspired by previous works. This allows us to transfer insight and intuition from these works directly our the neural system of interest, without reinventing the wheel. It is _because_ our model is related to existing generative models which allows for a deeper interpretation of hippocampal function. See general response section __Contributions__ for more on this point.
>
> > I anticipated grid like fields...
>
> Let us clarify: The MEC fields in fig 4b are a special case where we fixed HPC->MEC weights to the identity, hence why we see place-like fields in MEC (see line 520). In general our MEC representations are multimodal and grid-like (Fig S2a). Actually, they may be even more grid-like in larger environments when the attractor manifold would repeat itself due to the circular symmetry. We'll add a sentence to make this distinction clearer. Should the paper be accepted we'll be happy to add the temporal evolution of the weights as a figure to the supplement.
>
> > Inclusion of network prediction prior to learning...
>
> Thank you; we will make your proposed changes to the figures. Currently the y-axis in Fig 2b is log-base-10 meaning the $g$ prediction error decreases by 10x and $p$ by slightly less.
>
> > Path integration capability...
>
> This rate of error accumulation is only slightly worse that comparable models e.g. Sorcher et al. 2023, which reaches $\sim$10 cm error after 200 cm distance-travelled (=> 4 seconds at our agent's speed). Apples-to-apples comparisons are hard: we focus on small networks of $\sim$100 neurons and local learning rules which will always be outperformed by large networks (~4000 neurons) trained using backpropagation through time.
>
> Steps to improve our path integrator (which we'll mention in the paper) include
>
> * Gate some sensory data into soma during path integration, mirroring nature's lack of full sensory lesions (e.g. even in the absence of visual cues, here-unmodeled olfactory or tactile cues would persist).
> * Increase the number of neurons.
>
> > Application to a 2D navigation task...
>
> See general response section __2D environment__ for discussion and preliminary results.
>
> > ...vector-based navigation...
>
> We agree vector navigation merits further study; being tied to grid cells and continuous attractors, akin to those within our network. Regrettably, time constraints prevented including these extensions in our manuscript as we focused on novel results instead of replicating existing ones.
>
> > Insufficient ablation studies...
>
> Some ablation studies have already been performed to stress test the model
> * Lesioning of sensory inputs (Fig 3).
> * Relaxing the constraint that HPC receive unimodal spatial inputs (Fig. S2b)
> * Relaxing the identity constraint on weights from MEC to the conjunctive units (Fig. S2c)
>
> In the attached pdf you’ll find results for three new ablation studies, summarised in the general response section __Additional simulations__. These were performed in response to your review. Thank you for this valuable suggestion which has led to a geuinely interesting set of new results. We will include these (and any other ablation studies you feel important) in the supplement.
>
> > Would the plasticity rule work if it was purely Hebbian...
>
> No. The second term can be viewed as a normaliser or equally as the weight-dependent term in the “target” equation $U-V(w)$ (see Urbanczik, Neuron, 2014) pulling the dendritic voltage towards the somatic voltage until they are equal. Without it there is nothing to stop the weights from continuing to change/grow once they reach their target value of $U$ and blowing up. We’ll include a sentence to illuminate this intuition in the paper.
>
> > ...compare...to contrastive Hebbian learning?
>
> We aren't experts on CHL but were interested to read your comment and agree there are a number of intriguing similarities including alternative target-driven and non-target driven phases and two-term Hebbian-like learning rules. Without a more detailed mathematical analysis, which is arguably out of scope for the current paper, it is hard to say anything more strongly than this. Instead, may we suggest that we can flag these similarities in the paper for others to take note of and look forward to investigating this in future work as we likewise suspect there may be a deeper connection between the two.
>
> We hope our responses have addressed your inquiries and that the changes have enhanced the manuscript. If so, we'd be grateful if you'd consider revising your score to reflect this. If there are any specific actions we could take to further improve it, please let us know.

---

> > ### Comment · Reviewer_PiE7 · 2023-08-21
> >
> > Thank you for your response and running additional simulations. I still find the paper lacks certain clarity in motivating the novelty of the model architecture (why this architecture works even for different tasks using network analysis), justifying the seemingly high path integration error (Experiment with different network sizes could have been included to justify hypothesis). Athough, the abaltion studies are useful, I am inclined to keep the rating as it is (R6, C4).

---

> > > ### Author Response · Authors · 2023-08-21
> > >
> > > Thank you for your reply. Regarding clarity of the motivation and novelty please additionally consider our rebuttals to the other reviewers, particularly UjUg, where we describe numerous changes to the manuscript which will improve this aspect.
> > >
> > > Furthermore, as we explained we'd argue the error accumulation rate isn't much higher than other much larger and less biologically plausible models and falls in the ball park of comparable models trained with local learning rules. We remain confident that increasing $N$ and training time would monotonically improve performance as we see no intuitive reason for this not to be the case. We would be happy to add these experiments to a revised version of the manuscript.
> > >
> > > Please let us know whether any of your questions remain unanswered and whether these revisions might make you reconsider your rating.

---

> ### Author Response · Authors · 2023-08-15
> **Follow-up**
>
> Hi, just wanted to make sure you didn’t miss this. Let us know if you have any additional questions, we'll be happy to answer.

---

### Author Rebuttal · Authors · 2023-08-09

We thank all reviewers for their detailed and thoughtful comments and are glad they found it to be a "well written" and "logically presented" paper about a model which "could prove to be very important". We respond to each review individually but, for the benefit of all, here summarize three major aspects of our responses including additional experiments we have performed.

### Contributions

To reiterate our contribution: this paper substantially builds on an existing literature modelling the hippocampal formation's role in navigation (including path integration) and transferable structure learning. We take the novel approach of formulating it as a Helmholtz machine constructed from hierarchical layers of multi-compartmental neurons and show it can be "trained" with simple, local and biologically plausible Hebbian learning rules which can be approximately derived starting from an ELBO objective function.

Additionally, we show how the wake-sleep algorithm can be implemented by somatic gating between basal and apical inputs controlled by the hippocampal theta-oscillation. All in all our paper links deeply theoretical ideas about generative models (e.g. wake-sleep) to puzzling biological concepts (e.g. neural oscillations) in a manner we hope can seed fruitful discussion and progress.

We want to make clear that our intention to model HPC as a Helmholtz machine (as opposed to dreaming up or deriving an entirely new architecture) was a very concious choice. By not reinventing the wheel we can -- and hope the community will -- transfer insights from historical and recent ML research into generative models onto our understanding of the hippocampal formation. In doing so we avoid having to explain away lots of complexities with biologically implausible techniques such as back propagation as was done by others before us.

Our contribution should be of interest to both neuroscientists and ML'ers, for whom bioplausible implementations of key algorithms shed light on how the brain, and perhaps future AIs, can implement ML models in more data- and energy-efficient ways.

### Additional simulations

Reviewers suggested that additional ablation studies and experiments would strength our results. Here we summarise 3 additional experiments displayed in the attached pdf:

* __Removing plasticity from recurrent MEC synapses__. (Fig R1a)  The result is that learning fails and path integration does not emerge indicating recurrent plasticity is crucial and that interlayer plasticity alone cannot bypass the need for a tuned attractor manifold. As suggested by reviewers PiE7 and UjUg.
* __Removing plasticity from HPC $\leftrightarrow$ MEC synapses__ (Fig R1b) results in HPC never achieving internal consistency between basal and apical inputs. Despite this the model can still path integrate. The interpretation here is that HPC simply relays sensory information into MEC but cannot then “translate” MEC predictions back to the sensory code. This would be a problem for a real behaving agent who could not then easily combine sensory or internal predictions and would need two decoders to predict position (one for when the hippocampus is in wake mode, and another for sleep). As suggested by reviewers PiE7 and UjUg.
* __Increasing the synaptic noise 10x__ (Fig R1c & d) confirms that learning path integration is robust up to substantial amounts of noise (Fig R1c & d). As suggested by reviewer Lhyx.

We thank the reviewers for these suggestions leading to interesting new results which we will include in the manuscript.

### 2D environments

Three reviewers mentioned that our results would be more convincing in 2D. To test this __we performed some preliminary experiments in 2D__, with the results shown in the attached pdf. Now the agent moves in a 2D environment and has four sets of conjunctive cells in charge of controlling north, south, east and westward motion. Just like in 1D, after learning we observe the local "centre-surround" nature of the learned recurrent MEC synaptic connections (a hallmark feature of bump attractors, Fig R2b left), and also that the synapses from the four sets of conjunctive cells are each skewed in four opposite directions (a hallmark feature of path integration, Fig R2b right). Fig R2c shows path integration where every 1 second sensory input is provided to recalibrate the 2D bump attractor. In between these instances the activity bump remains stable (albeit only briefly).  This result indicates that path integration abilities are beginning to emerge in the 2D model. Note that, due to computational and time constraints, this model has not been tuned nor trained for the required amount of time and therefore it is likely we are not seeing optimal performance of the bump attractor.

To be clear, we do not intend these results as a comprehensive exploration of 2D and recognise they leave many questions unanswered. Unfortunately 2D takes substantially longer to simulate and tune due to the increased complexities of motion modelling and the much larger number of cells required. For these reasons, properly exploring 2D remains out of scope and we choose to focus only on 1D as the simplest setting where all core features of path integration can be tested comprehensively. We include them here as they give us some confidence that this approach would scale to 2 - and potentially higher - dimensions.

---

### Decision · Program_Chairs · 2023-09-21

**Decision:**

Accept (poster)

**Comment:**

This paper presents an abstract model of hippocampal learning based on a sequential version of the Helmholtz machine. The authors show that model accurately infers the latent state of high-dimensional sensory environments and generates realistic predictions. They also show that it can be used for navigation and transfer between environments (using relatively toy environments).

Overall, the reviewers had concerns related to the novelty of the paper, the clarity of what it contributes beyond previous work, and the simplicity of the environments used. After a fairly extensive set of rebuttal discussions, 3/4 reviewers felt that the paper met the bar for acceptance. As such, and accept decision was reached.